# Cellular interpretation of the long-range gradient of Four-jointed activity in the *Drosophila* wing

**Rosalind Hale[1,2], Amy L Brittle[1,2†], Katherine H Fisher[1,2†], Nicholas A M Monk[3], David Strutt[1,2]***

[1]Bateson Centre, University of Sheffield, Sheffield, United Kingdom; [2]Department of Biomedical Science, University of Sheffield, Sheffield, United Kingdom; [3]School of Mathematics and Statistics, University of Sheffield, Sheffield, United Kingdom

**Abstract** To understand how long-range patterning gradients are interpreted at the cellular level, we investigate how a gradient of expression of the Four-jointed kinase specifies planar polarised distributions of the cadherins Fat and Dachsous in the *Drosophila* wing. We use computational modelling to test different scenarios for how Four-jointed might act and test the model predictions by employing fluorescence recovery after photobleaching as an in vivo assay to measure the influence of Four-jointed on Fat-Dachsous binding. We demonstrate that in vivo, Four-jointed acts both on Fat to promote its binding to Dachsous and on Dachsous to inhibit its binding to Fat, with a bias towards a stronger effect on Fat. Overall, we show that opposing gradients of Fat and Dachsous phosphorylation are sufficient to explain the observed pattern of Fat–Dachsous binding and planar polarisation across the wing, and thus demonstrate the mechanism by which a long-range gradient is interpreted.

**\*For correspondence:** d.strutt@
sheffield.ac.uk

[†]These authors contributed equally to this work

## Introduction

Planar polarity (also known as planar cell polarity [PCP]) refers to the coordinated polarisation of cells within the plane of a tissue such as an epithelium. How epithelia are planar polarised and how planar polarisation is co-ordinated across a tissue has intrigued researchers for decades. In the late 1950's, Locke presented evidence that orienting gradients could control epithelial tissue patterning (*Locke, 1959*), but despite years of research, the mechanisms by which graded cues might mediate the coordinated polarisation of individual cells remain incompletely understood (for review see *Strutt, 2009*).

Of the known molecular systems regulating planar polarity, only for the *Drosophila* Fat-Dachsous-Four-jointed (Ft-Ds-Fj) pathway is there strong evidence for a primary role of graded activity in providing orienting cues (*Zeidler et al., 1999*, *2000*; *Casal et al., 2002*; *Strutt and Strutt, 2002*; *Yang et al., 2002*; *Ma et al., 2003*; *Matakatsu and Blair, 2004*; *Simon, 2004*; *Ambegaonkar et al., 2012*; *Brittle et al., 2012*). Ft and Ds are large atypical cadherins known to bind to each other heterophilically (*Ma et al., 2003*; *Matakatsu and Blair, 2004*), and Fj is a kinase shown to be active in the Golgi (*Strutt et al., 2004*; *Ishikawa et al., 2008*). Complementary graded expression patterns of Ds and Fj (*Zeidler et al., 1999*, *2000*; *Yang et al., 2002*; *Ma et al., 2003*) result in the planar polarisation of Ft and Ds across cells, with (in the developing wing) Ds accumulating distally and Ft accumulating proximally (*Ambegaonkar et al., 2012*; *Bosveld et al., 2012*; *Brittle et al., 2012*). How the complementary expression patterns of Fj and Ds result in the accumulation of Ft and Ds on opposing cell edges is still under investigation, however, the data suggest that higher proximal expression of Ds and the opposing gradient of Fj expression leads to a gradient of Ft–Ds dimer formation (*Casal et al., 2006*; *Lawrence et al., 2008*; *Strutt, 2009*).

**eLife digest** Epithelial cells form sheets that line the body surfaces and internal cavities of animals—such as the skin and the lining of the gut. Certain structures on the surface of epithelial cell sheets—for example scales, hair, and feathers—are often all orientated in a particular direction. Epithelial cells with structures organised like this are described as being 'planar polarised'.

Different proteins work together to set up planar polarity in a sheet of epithelial cells. Dachsous and Fat are two proteins that are found in the cell membranes of epithelial cells, including in the wings of the fruit fly *Drosophila*. These proteins bind to each other and link a cell to its neighbour. Dachsous and Fat accumulate on opposing sides of each cell: Fat accumulates on the side closest to the fly's body, and Dachsous builds up on the side closest to the wing tip. This pattern provides directional cues that help orientate surface structures, and the pattern is established, in part, by the activity of an enzyme called Four-jointed.

Four-jointed adds phosphate groups onto Dachsous and Fat. The activity of the Four-jointed enzyme forms a gradient along a developing wing: levels are low near the fly's body, and high at the wing tip. Previous experiments performed on cells grown in the laboratory showed that when Four-jointed adds phosphate groups to Fat and Dachsous, it prevents Dachsous from binding to Fat. However, it also makes Fat more able to bind to Dachsous. These opposing effects are thought to cause the proteins to accumulate on opposing sides of each cell. However, this has yet to be demonstrated in real tissue, not least because of the technical difficulty of measuring whether Fat-Dachsous binding has occurred in living organisms.

Here, Hale et al. overcome this challenge using a method called 'fluorescence recovery after photobleaching' (or FRAP) to measure Fat and Dachsous binding in the epithelial cells in the developing *Drosophila* wing. Combining these experimental results with a computational model confirmed the findings of previous laboratory studies: that Four-jointed makes it easier for Fat to bind to Dachsous, and makes it more difficult for Dachsous to bind to Fat. The opposing effects on the activity of Fat and Dachsous that result from the Four-jointed gradient in the developing wing are able to fully explain the observed patterns of Fat-Dachsous binding and of planar polarisation across the wing.

Overall, Hale et al. demonstrate how a gradient of protein activity that spans many cells is sensed and interpreted by individual cells to establish planar polarity. However, exactly how the phosphate groups added to Dachsous and Fat by Four-jointed modifies how they bind to each other remains a question for future work.

*Waddington (1943)* first reported a genetic interaction between *fj* and *ds*, and subsequent work revealed that Fj was able to regulate the localisation and function of both Ft and Ds (*Casal et al., 2002*; *Strutt and Strutt, 2002*; *Yang et al., 2002*; *Ma et al., 2003*; *Cho and Irvine, 2004*; *Strutt et al., 2004*; *Casal et al., 2006*). Fj is able to phosphorylate several cadherin repeats of both Ft and Ds (*Ishikawa et al., 2008*), raising the possibility that this is a mechanism by which Fj regulates Ft–Ds binding. This led to examination of the result of Fj phosphorylation on Ft and Ds binding in vitro (*Brittle et al., 2010*; *Simon et al., 2010*).

Using a cell aggregation assay, based on *Drosophila* S2 cells co-transfected with Fj, Ds, or Ft, *Brittle et al. (2010)* deduced that Fj was able to act on three previously identified serines in Ds to inhibit the binding of Ds to Ft (*Ishikawa et al., 2008*; *Brittle et al., 2010*). Using an alkaline phosphatase-based cell surface binding assay, *Simon et al. (2010)* similarly found that Fj inhibited the ability of Ds to bind to Ft and were also able to demonstrate an improvement in Ft binding to Ds. In addition, in vivo assays examining the propagation of polarity from over-expression clones also found that over-expressed Fj promoted Ft activity and inhibited Ds activity (*Brittle et al., 2010*).

So far there is no direct in vivo evidence that Fj normally does act on both Ft and Ds to modulate their binding during tissue patterning, or that any such activity is directly dependent on the mapped phosphorylation sites in the cadherin domains. Furthermore, it is unclear how the observed asymmetric subcellular distributions of Ft and Ds are related to the proposed differences in binding affinities between neighbouring cells and across the tissue.

To address these issues, we have carried out in vivo studies of Ft and Ds behaviour, making use of both normal protein forms and variants mutated at the mapped phosphorylation sites, and using protein mobility as measured using fluorescence recovery after photobleaching (FRAP) as a novel assay for in vivo binding activity. Furthermore, to better understand the possible effects of Fj phosphorylation on Ft–Ds binding and how this might lead to their asymmetric subcellular distributions, we have combined our experimental approach with computational modelling.

## Results

### Modelling Ft and Ds interactions and the emergence of cellular asymmetry

As an aid to understand the possible consequences for the generation of cellular asymmetry of Fj acting on either Ft or Ds or both, we generated a computational model reflecting a one-dimensional line of cells each with two compartments, to simulate Ft–Ds binding between cells according to a Fj gradient (see 'Materials and methods'). In the model, Fj is allowed to phosphorylate either Ft or Ds in proportion to its concentration, and Ft and Ds are then permitted to freely bind at cell edges until they reach equilibrium (see *Figure 1A*). To produce a representative model, we measured the gradient of Fj expression across *Drosophila* larval wing discs (*Figure 1C*), using *fj* null clones to determine the appropriate background correction (*Figure 1C''*). A gradient of 2.5–3.5% between adjacent cells (*Figure 1B*) was observed along the proximo-distal axis. We also allowed redistribution of Ft and Ds to occur in the model, in accordance with our experimental observations of protein mobility (see *Figure 2C,D,L*). A hierarchy of binding affinities was used, based on the in vitro experiments (*Brittle et al., 2010*; *Simon et al., 2010*), with phosphorylated Ft (FtP) and unphosphorylated Ds (Ds) producing the strongest partnership, and unphosphorylated Ft (Ft) and phosphorylated Ds (DsP) producing the weakest partnership (see 'Materials and methods').

Using these starting parameters, our modelling predicts that even if Fj acts on only one of either Ft or Ds, then a weak asymmetry of both Ft and Ds is produced across cells (*Figure 1D,D',E,E'*). However, in both cases there is also a shallow gradient of bound protein across the tissue. If, however, Fj acts on both Ft and Ds, there is improved cellular asymmetry and a negligible tissue gradient (*Figure 1D'',E''*). Notably, neither situation predicts the approximate two-fold asymmetry in Ds distribution across the cell axis that is experimentally observed (*Brittle et al., 2012*, see 'Discussion').

Thus, in our model, Fj acting on only Ft, or only Ds, or both results in asymmetric subcellular distributions of Ft and Ds. Therefore, in order to resolve the role of Fj in Ft–Ds asymmetry generation, we sought to understand its effects on Ft–Ds binding in vivo.

### Ft and Ds exhibit stable populations at the cell junctions

As there is no method for directly measuring the strength of binding of two proteins in the in vivo context of a living tissue, we instead employed FRAP to assay the mobility of Ft and Ds in the developing *Drosophila* wing. During FRAP, flies with an EGFP-tagged protein (e.g., Ds-EGFP) are used, and a fluorescent region of interest is selected and bleached. The same region is imaged over time and recovery of fluorescent protein is measured. Recovery is due to movement of unbleached protein into the bleached area from elsewhere in the cell and any protein that has been bleached remains bleached and is not reactivated. Therefore, if after bleaching and imaging, over time we see, for example, fluorescence recovering to 30% of the starting intensity we assume that 70% of the bleached protein remains and is therefore stably bound at the junction. We perform all of our experiments using the same microscope settings (unless otherwise stated) and since protein levels (and thus fluorescent intensity) can vary between genotypes, we must take this into account during our analysis. To do this, we multiply the starting intensity by the stable fraction to get a final stable amount (see *Figure 2—figure supplement 1*). We make the assumption that a decrease in the speed or size of the mobile population provides a readout of increased binding interactions. For instance, if the size of the mobile population (unstable fraction) increases in a particular genotype and, once the intensity is taken into account, a smaller stable amount of overall protein is calculated, we assume that binding affinity has decreased. Additionally, if the speed of protein recovery after bleaching increases in a particular mutant, we infer that binding is not as strong as 'wild-type' and the protein can therefore relocate more quickly. FRAP was performed as previously described (*Strutt et al., 2011* see also 'Materials and methods' and *Figure 2—figure supplement 1*: individual data points and

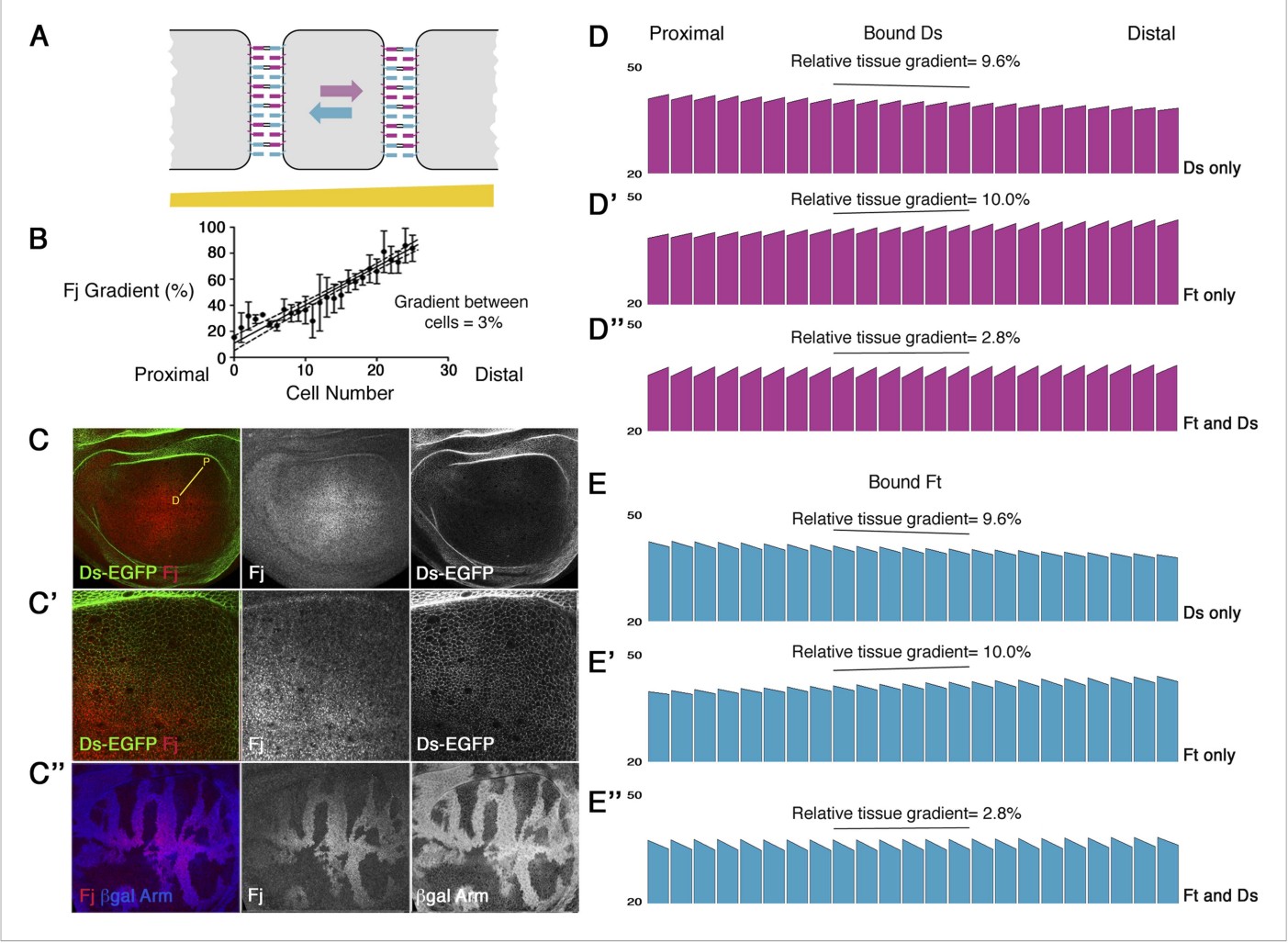

**Figure 1**. Computational modelling of the effect of a Fj gradient on Ft–Ds binding. (**A**) Cartoon illustrating Ft–Ds binding and the predicted effect of a Fj gradient on the distribution of Ft–Ds heterodimers in a single cell. In the absence of any gradients (i.e., uniform Ft, Ds, and Fj expression across the tissue), Ft (blue) and Ds (purple) freely associate at junctions, with a certain proportion binding to form heterodimers (indicated by double black bars). We predict that by adding a Fj gradient (yellow bar) an asymmetric distribution of Ft–Ds binding across a cell will occur. If Fj inhibits Ds binding and promotes Ft binding more strongly to the right, Ft molecules in each cell prefer to bind to Ds in the next left-most cell (with which they have a stronger binding interaction), and so preferentially accumulate at left cell edges; similarly Ds molecules prefer to associate with Ft in the next right-most cell and in turn accumulate at right cell edges. Hence, the overall consequence of a Fj gradient and free movement of Ft and Ds within cells is the generation of cellular asymmetries of Ft and Ds distributions. (**B**) Graph illustrating the proximal-distal measurements of the Fj gradient across *Drosophila* third instar larval wing discs, showing a typical gradient of around 3% (dashed lines indicate 95% confidence intervals, error bars indicate standard deviation (SD), n = 5). Cell number increases from 0 to 30 moving from proximal to distal. (**C**) Confocal images of wing discs from animals with EGFP knocked-in to the endogenous *ds* locus, stained for Fj (red) and observed for native GFP fluorescence (green). A yellow line indicates where Fj gradient measurements were taken from proximal [P] to distal [D]. (**C'**) An example zoomed-in image showing the distribution of Fj and Ds in this region. (**C"**) *fj*^*d1* mutant clones (absence of blue ß-gal staining, cell junctions also labelled in blue) demonstrate that Fj is expressed throughout the tissue. Mutant clones were used to subtract background from the gradient measurements. (**D** and **E**) A computational model of Ft–Ds binding at cellular junctions. Each bar represents one cell and the scale indicates the amount of bound Ds or Ft in arbitrary units at junctions between cells (note scale is cut off at 20 units), also see *Figure 1—figure supplement 1*. Sloped top edges represent the difference between proximal and distal cell edges within one cell. A more graded slope within a cell indicates an increase in asymmetry. 23 cells are represented and the observed Fj gradient of 3% (**B**) has been used. Looking at both bound Ds (**D**) and bound Ft (**E**) shows Ds is preferentially localised distally and Ft proximally. If Fj acts on Ds only (**D** and **E**), weak asymmetry is seen within cells (3.8% increase) and a tissue gradient of higher to lower Ft–Ds binding is seen as Fj increases. If Fj acts on Ft only (**D'** and **E'**) similar weak asymmetry (3.6% increase) is seen within cells, however, a tissue gradient of lower to higher Ft–Ds binding is observed as Fj increases. If Fj can act on both Ft and Ds (**D"** and **E"**) stronger asymmetry is seen within cells (7.9% increase) and the tissue gradient is much reduced.

The following figure supplement is available for figure 1:

**Figure supplement 1**. Representation of mathematical model.

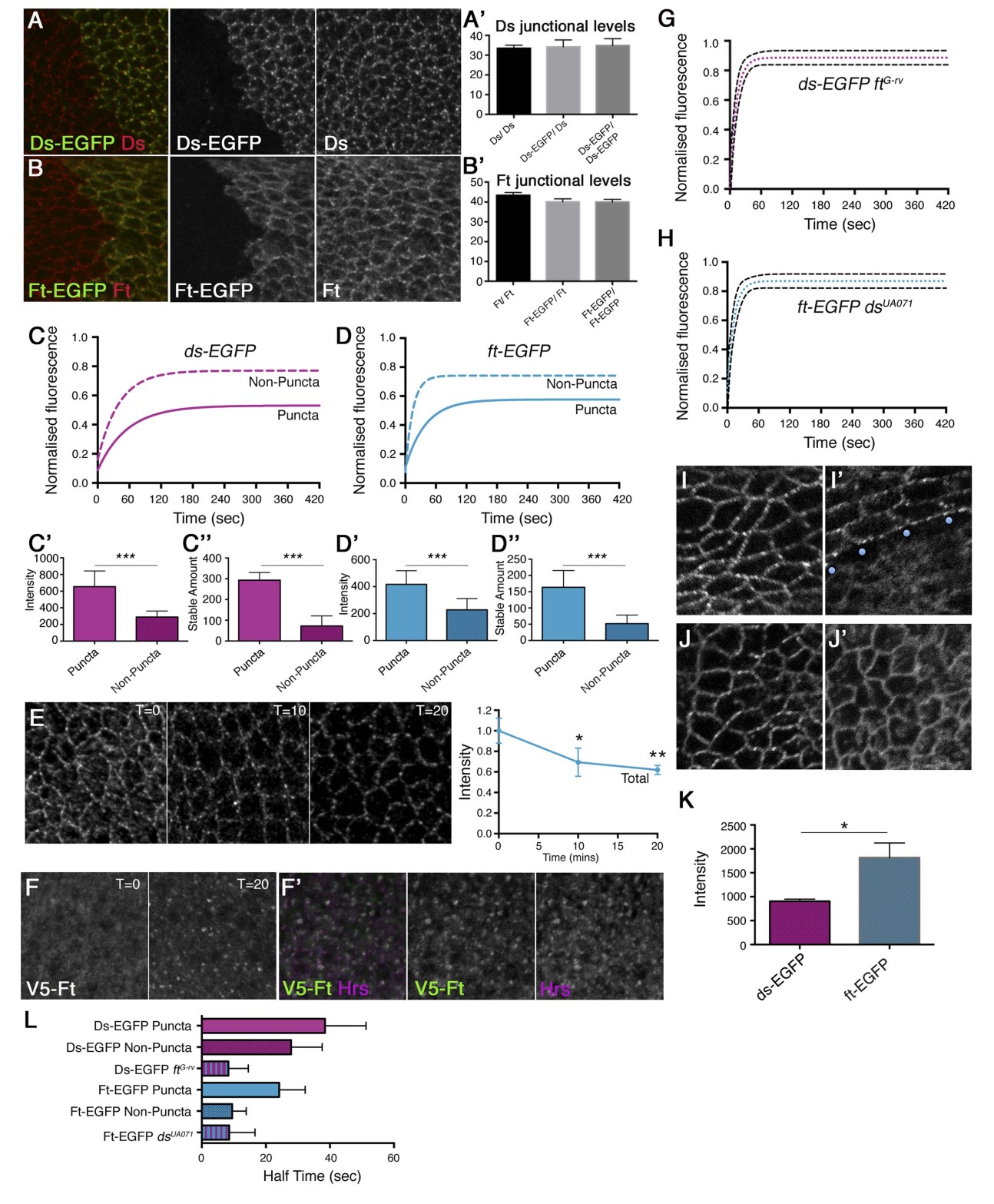

**Figure 2**. Ft and Ds show mutually dependent stable fractions at junctions. Wing discs containing clones of (**A**) *ds-EGFP/+* and (**B**) *ft-EGFP/+* labelled for EGFP (green) and Ds or Ft, respectively (red), show that addition of an EGFP tag does not affect protein localisation. Quantification of (**A'**) Ds or (**B'**) Ft junctional protein confirms expression levels of tagged proteins are similar to wild-type. FRAP analysis of (**C**) Ds-EGFP and (**D**) Ft-EGFP puncta (bright regions, solid line) and non-puncta (dashed line) reveals a large stable fraction of protein in puncta (see *Figure 2—figure supplement 1* for an example of

*Figure 2. continued on next page*

*Figure 2. Continued*

a puncta and non-puncta region and *Figure 2—figure supplement 2* for FRAP graphs with individual data points). Multiplying the intensity of FRAP regions prior to bleaching (**C′** and **D′**) by the stable fraction plateau (**C** and **D**) reveals a final 'stable amount' of protein at junctions (**C″** and **D″**) and a large significant difference between puncta and non-puncta regions (both p = 0.0001, error bars denote standard error (SEM) here and in all remaining figures). (**E**) Apical confocal sections of a V5-Ft antibody internalisation assay in pre-pupal wing, with time points of 0, 10, and 20 min, reveal persistent Ft puncta over time while quantification of total apical levels (right panel) shows protein internalisation is occurring (p = 0.01 at 10 min and p = 0.001 at 20 min) (**F**) Sub-apical confocal sections of pre-pupal wing show V5-Ft previously at the cell surface is seen in discrete intracellular puncta at t = 20, where it co-localises with early endosome marker Hrs (magenta) (**F′**), confirming antibody internalisation rather than antibody drop off. FRAP analysis of Ds-EGFP in $ft^{G-rv}$ clones (**G**) and Ft-EGFP in a $ds^{UA071}$ background (**H**) resulted in more rapid and increased protein recovery. Dashed lines denote 95% confidence intervals. As recovery did not reach 100%, a small stable population could remain. Also see *Figure 2—figure supplement 4* for FRAP graphs with individual data points. Re-bleaching experiments suggest that this is not an experimental artefact (see *Figure 2—figure supplement 5*). Live images of Ds-EGFP (**I**) and Ds-EGFP with a $ft^{G-rv}$ clone (**I′**; blue dots indicate first row of mutant cells) show a loss of puncta and diffuse distribution of Ds-EGFP when Ft is not present. Similar distribution of Ft-EGFP (**J**) is seen in a $ds^{UA071}$ mutant (**J′**). (**K**) Comparison of Ds-EGFP and Ft-EGFP protein levels taken at the same confocal settings suggests there is almost twice as much Ft-EGFP (n = 4 wings) as Ds-EGFP (n = 2 wings) in puncta regions (p = 0.01). Intensity measurements were taken from manually selected puncta using 1 μm² ROIs. Around 50 ROI measurements were taken per wing and average intensities were plotted and analysed using an unpaired t-test. (**L**) Bar chart representing time taken for 50% protein recovery to occur during FRAP experiments. Both Ft-EGFP and Ds-EGFP mobile fractions exhibit slower recovery in puncta compared to non-puncta and mutants.

The following figure supplements are available for figure 2:

**Figure supplement 1**. Description of FRAP method.

**Figure supplement 2**. Individual data points for FRAP experiments.

**Figure supplement 3**. FRAP on Ds-EGFP in the pre-pupal wing.

**Figure supplement 4**. Individual data points for FRAP experiments.

**Figure supplement 5**. Control re-bleach FRAP experiments.

confidence intervals are also provided in the relevant figure supplements and *Supplementary file 1*) in the same proximal region of the wing disc (unless otherwise stated) using a strain of flies expressing the Ds protein endogenously tagged with EGFP (*Brittle et al., 2012*), and a newly generated strain in which EGFP was inserted at the C-terminus of the endogenous Ft coding region (see 'Materials and methods'). Both transgenes were found to be expressed and localised similarly to wild-type (*Figure 2A,B*) and insertion of the tag did not affect wing size or shape suggesting protein function is normal.

Live imaging of both Ft-EGFP and Ds-EGFP (*Figure 2I,J*) revealed that the junctional populations exhibit a punctate distribution, as previously seen by immunolabelling in fixed tissue (*Ma et al., 2003*; *Brittle et al., 2012*) and also for components of the 'core' planar polarity pathway (*Strutt et al., 2011*). In our FRAP experiments, puncta and non-puncta regions were bleached and fluorescence recovery was measured over time (*Figure 2C,D*, *Figure 2—figure supplement 2*). The Ft-EGFP signal was almost twice the level of Ds-EGFP (*Figure 2K*), therefore laser power had to be adjusted accordingly between experiments meaning the final stable amounts for Ds-EGFP cannot be compared directly to those for Ft-EGFP. For both puncta and non-puncta, there was some recovery of fluorescence demonstrating that the proteins were mobile at junctions. Recovery was not complete indicating that there was also a population of stable Ds-EGFP and Ft-EGFP that did not recover during the time period used. Measuring protein recovery revealed a significant difference in stability between puncta and non-puncta regions (*Figure 2C,D* and *Table 1*) with puncta showing an increased amount of stable protein (*Figure 2C″,D″*, both p = 0.0001) when taking pre-bleach intensity levels into account (*Figure 2C′,D′*). Overall, the FRAP assays indicated that there were both stable and unstable populations of Ft-EGFP and Ds-EGFP present at junctions, with stable material concentrated into bright punctate regions.

As an independent assay to confirm the presence of stable and unstable protein populations, we performed an antibody internalisation assay in the pre-pupal wing (see 'Materials and methods' and

**Table 1.** Comparison of rate and stabilisation data

| Genotype | Stable amount | SEM of stable amount | Half time | Confidence interval |
|---|---|---|---|---|
| Ds-EGFP Homozygous Puncta | 292.9 | ±18.5 | 38.5 | 30.8 to 51.3 |
| Ds-EGFP Homozygous Non-Puncta | 72.1 | ±21.8 | 27.9 | 22.13 to 37.6 |
| Ds-EGFP $ft^{G-rv}$ Homozygous | n/a | n/a | 8.3 | 5.8 to 14.5 |
| Ds-EGFP $fj^{d1}$ Homozygous Puncta | 167.2 | ±21.6 | 33.89 | 25.5 to 50.5 |
| Ds-EGFP $fj^{d1}$ Homozygous Non-Puncta | 26.4 | ±20.5 | 5.6 | 3.8 to 10.3 |
| Ft-EGFP Homozygous Puncta | 148.0 | ±9.5 | 24.2 | 19.28 to 32.3 |
| Ft-EGFP Homozygous Non-Puncta | 52.0 | ±10.6 | 9.5 | 7.2 to 13.9 |
| Ft-EGFP $ds^{UA071}$ Homozygous | n/a | n/a | 8.5 | 5.7 to 16.6 |
| Ft-EGFP $fj^{d1}$ Homozygous Puncta | 114.1 | ±1.4 | 16.8 | 13.3 to 22.8 |
| Ft-EGFP $fj^{d1}$ Homozygous Non-Puncta | 52.6 | ±19.0 | 11.4 | 8.7 to 16.6 |
| Ds-EGFP Homozygous Puncta Distal (High Fj) | 378.0 | ±17.6 | 93.0 | 59.1 to 217.6 |
| Ds-EGFP $fj^{d1}$ Homozygous Puncta Distal | 142.7 | ±28 | 69.7 | 51.3 to 108.6 |

*Strutt et al., 2011*) to assess the endocytic turnover of Ft, using a Ft transgene tagged extracellularly with a V5 epitope (*Feng and Irvine, 2009*). The internalisation assay is performed in pre-pupal wings as the peripodial membrane prevents the assay from being effective in the wing disc. Live pre-pupal wings were dissected in Schneider's medium and incubated with antibody against V5 at 0°C followed by washing and chasing at room temperature before fixation. The amount of V5-Ft at the cell surface was determined via incubation with secondary antibody in the absence of detergent. When endocytosis was allowed by moving the tissue to room temperature the total amount of apical cell surface protein decreased over time suggesting protein internalisation was occurring and indicating the presence of an unstable cell surface population of V5-Ft (*Figure 2E*). To confirm that protein loss was not due to antibody drop off, V5-Ft previously found at the surface was traced to internal vesicles (*Figure 2F*) and found to co-localise with the early endosome marker Hrs (*Figure 2F'*). Populations of V5-Ft appeared to be resistant to endocytic turnover with persistent puncta of cell surface V5-Ft observed after 20 min (*Figure 2E*). This was consistent with the puncta stability observed in FRAP experiments both in wing discs (*Figure 2D''*) and also in the pre-pupal wing (*Figure 2—figure supplement 3*). Note however, that although we assume that these remaining populations in each assay are the same, we cannot be certain that this is the case. During FRAP, we are unable to observe the fate of the bleached protein. Furthermore, in the internalisation assay, protein might be internalised and return to the same sites via exocytosis.

## Ft and Ds require each other for junctional stability

To understand whether the observed protein stability at junctions was due to the presence of heterophilic binding between Ft and Ds, FRAP was performed on both Ft-EGFP and Ds-EGFP in *ds* and *ft* null mutant backgrounds, respectively (*Figure 2G–J*, *Figure 2—figure supplement 4*). Removal of the putative binding partner resulted in the loss of obvious puncta and the almost complete recovery of fluorescence after bleaching, indicating that the majority of protein had become unstable at junctions. Thus, the observed stability of each protein is dependent on the presence of the binding partner.

Comparing data for the rate of recovery of fluorescence of Ds-EGFP (*Table 1* and *Figure 2L*) further revealed that the mobile populations of Ds-EGFP show slower movement, either in puncta or non-puncta, in the presence of Ft, compared to that in the absence of Ft (where no puncta are observed). This suggests that in addition to the immobile population of Ds bound to Ft which is concentrated in puncta as described in the previous section, there is also a mobile population of Ds-EGFP present in both puncta and non-puncta regions which is bound to Ft and shows less rapid movement than free Ds-EGFP (i.e., Ds-EGFP in the absence of Ft). The reduction in the rate of

movement of these mobile Ds-Ft heterodimers in puncta as opposed to non-puncta might indicate the presence of a cis-dimerisation mechanism that promotes the clustering of heterodimers into the puncta. Alternatively, mechanisms such as physical interactions with the cytoskeleton similar to those seen for E-Cadherin clustering (*Cavey et al., 2008*) may be responsible for the reduced mobility.

Our data on the rate of recovery of Ft-EGFP (*Table 1* and *Figure 2L*) also support these conclusions. However, in this case there is a negligible difference in the rate of movement of Ft-EGFP in non-punctate regions in the presence of Ds, as compared to the rate of movement of Ft-EGFP in the absence of Ds, and also a faster rate of Ft-EGFP mobility within puncta as compared to Ds-EGFP in puncta. We surmise that these differences are due to there being an excess of Ft-EGFP present over Ds (see *Figure 2K*), resulting in an increased proportion of the population of Ft-EGFP being unbound and thereby free to move into bleached regions.

Overall, we conclude from our FRAP experiments using 'wild-type' Ft-EGFP and Ds-EGFP, that Ft and Ds bind to each other across cell membranes resulting in the production of immobile fractions of Ft and Ds at cell junctions, and a reduction in mobility of Ft and Ds at junctions.

## The in vivo effect of Fj on Ft/Ds stability

Having thus demonstrated that Ft–Ds binding can be measured in vivo by virtue of the effects that binding has on protein mobility and stability, we next sought to understand the effects of Fj on Ft–Ds binding. Fj is able to phosphorylate both Ft and Ds (*Ishikawa et al., 2008*) and modify the binding affinities between the two proteins in vitro (*Brittle et al., 2010*; *Simon et al., 2010*); however, the functional in vivo result of these modifications is unknown. To investigate this, we performed FRAP on endogenously expressed Ft-EGFP and Ds-EGFP in a *fj* mutant background, measuring fluorescence recovery to infer the stability of Ft–Ds dimers (*Figure 3*, *Figure 3—figure supplement 1*). As Ft–Ds binding across junctions is mutually dependent, we infer that if one protein gains or loses stability, so will the other. Moreover, as Fj is thought to act on Ft and Ds in opposite ways, if we remove Fj, we might not expect to see any difference in overall stability of Ft–Ds dimers as the positive and negative effects could cancel each other out. If a difference is observed, it suggests that phosphorylating one protein has a stronger effect than phosphorylating the other.

The co-expression of Fj with Ds in a cell culture assay has been shown to reduce the ability of Ds to bind to Ft (*Brittle et al., 2010*). Therefore, we hypothesise that removing Fj might increase the stability of any Ft–Ds dimer (corresponding to a decrease in the height of the FRAP recovery plateau). However, we found that Ds-EGFP became less stable upon removal of Fj (*Figure 3A″*, p = 0.004), with a reduction in both the stable fraction (i.e., an increase in the height of the plateau) and the stable amount at junctions (*Figure 3A,A″*). Stills of live FRAP experiments demonstrate the increased recovery of Ds-EGFP in a *fj* mutant background after 70 s (*Figure 3C*). Co-expression of Fj with Ft has been shown to increase the affinity of Ft for Ds (*Simon et al., 2010*); therefore, we should expect a loss of stability of Ft-EGFP in a *fj* background. In this case analysis of Ft-EGFP recovery in a *fj* background did result in a reduction in stable fraction and stable amount at junctions (*Figure 3B,B″*, p = 0.04). The half time rate of recovery of Ft-EGFP and Ds-EGFP was also decreased in the absence of Fj, consistent with a reduction of binding affinity allowing more rapid mobility (*Table 1*).

Thus, the overall result of removing Fj was a reduction in stability of the Ft–Ds dimer. A loss of stability, rather than a gain, suggests that, at least in the wing disc, the effect Fj has on Ft is dominant to any which it might have on Ds.

## Fj acts on both Ft and Ds in vivo altering Ft/Ds dimer stability

Using endogenously tagged Ft and Ds proteins, we have uncovered a dominant effect of Fj on Ft; however, the overall result may mask any potential consequence of Fj action on Ds. As the in vitro evidence suggests that Fj modifies Ft–Ds binding via phosphorylation of their extracellular cadherin repeats (*Ishikawa et al., 2008*; *Brittle et al., 2010*; *Simon et al., 2010*), we turned to mutations in the mapped phosphorylation sites to separate effects on each molecule. We have previously described fly strains expressing EGFP-tagged Ds phosphorylation mutants and mimetics uniformly under control of the *Actin5C* promoter (*Brittle et al., 2010*). Under these expression conditions, protein levels are higher than endogenous and there is a loss of visible puncta meaning that FRAP experiments are performed solely in junctions and puncta and non-puncta are not distinguished. Microscope settings were also altered to take the increased intensity levels into account. All of the phosphomutant

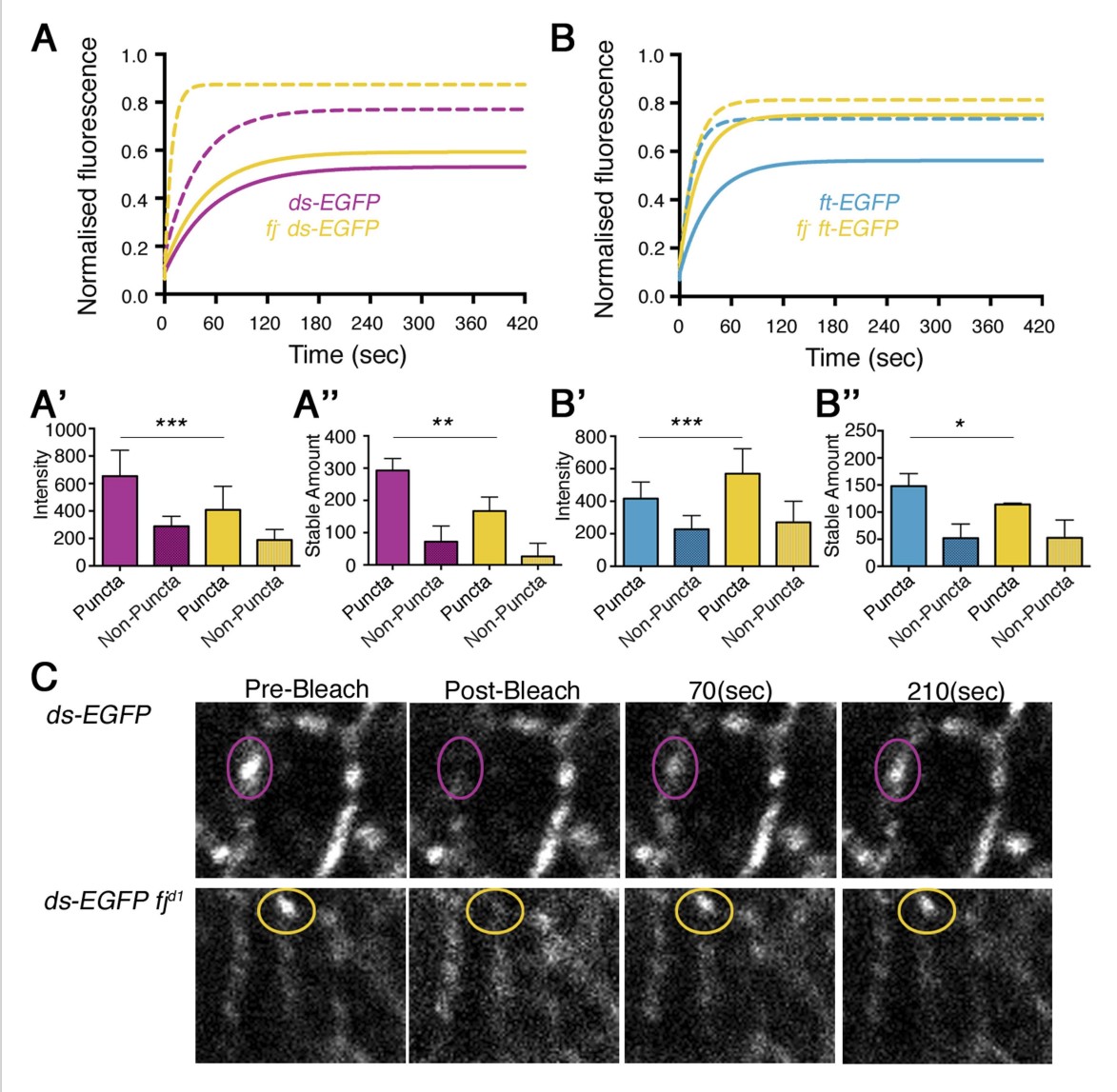

**Figure 3**. Ft and Ds stability upon loss of Fj. Stable fraction of (**A**) Ds-EGFP and (**B**) Ft-EGFP in a *fj^{d1}* background (yellow lines). Solid lines represent puncta and dashed lines represent non-puncta. 'Wild-type' values are plotted for comparison. Intensity of Ds-EGFP decreases (**A'**) and Ft-EGFP increases (**B'**), in a *fj^{d1}* background, whereas stable amount of protein in puncta falls in both (**A''** (p = 0.004) and **B''** (p = 0.04)). The reduction in Ds-EGFP levels at junctions in the absence of Fj is consistent with less Ds being bound to Ft in this situation and excess unbound Ds being removed from junctions. The reason for an increase in Ft levels is unknown, as presumably less Ft is bound to Ds, but suggests that unbound Ft is not removed from junctions. See *Figure 3—figure supplement 1* for FRAP curves with individual data points. (**C**) Live images of wing discs taken during a FRAP experiment show the recovery of puncta over time in *ds-EGFP* and *ds-EGFP* in a *fj^{d1}* background. Protein recovery after 70(sec) is increased in a *fj^{d1}* background and overall protein levels appear reduced.

The following figure supplement is available for figure 3:

**Figure supplement 1**. Individual data points for FRAP experiments.

experiments were performed in relevant mutant backgrounds so the *Actin5C* construct was the only form of expression of the protein in question. For example, if we were looking at Act-Ds-EGFP, we performed the experiment in a *ds* background.

If Fj does have an independent effect on Ds, we might expect that preventing Fj from phosphorylating Ds (Ds phosphomutant) whilst not affecting Ft phosphorylation (by leaving Fj cellular

activity intact) should allow us to see this. Based on previous experiments where phosphorylating Ds reduces the binding affinity between Ft and Ds (*Brittle et al., 2010*), we are able to hypothesise that mutating the phosphorylation sites in Ds, thus preventing Fj phosphorylation, should improve binding affinity and therefore increase the stable amount of bound protein. We also hypothesise that mutating the phosphorylation sites of Ft (so Ft cannot be phosphorylated by Fj) will result in a reduction in binding affinity and therefore a reduced stable amount.

When comparing the stability of 'wild-type' Ds-EGFP (expressed from the *Act-ds-EGFP* transgene) and a form of Ds with the phosphorylation sites mutated to alanine to block phosphorylation (*Act-ds-$^{S>Ax3}$-EGFP*; *Figure 4A*, *Figure 4—figure supplement 1*), there was a strong increase in the amount of stable protein present in the mutant (*Figure 4A″*, 196.3 $\pm$ 18 and 428.9 $\pm$ 35 intensity units respectively, p = 0.0004). As Fj was still expressed and presumably able to act normally on Ft in this experiment, the increase in stable amount must be solely due to the inability of Fj to phosphorylate Ds. Additional loss of Fj, to remove potential Ft phosphorylation from the experiment, resulted in the expected decrease in stability, back to a level similar to 'wild-type' (*Figure 4B,E*, *Figure 4—figure supplement 1*), again consistent with the effect of Fj on Ft being dominant to that on Ds. A bar chart representing all of the different stable amounts of *Act-DsEGFP* is provided in *Figure 4E*.

We were unable to detect any decrease in stability using our previously described Ds phosphomimetic (*Act-ds-$^{S>Dx3}$-EGFP*) fly strain (*Figure 4—figure supplement 1*). We suspect that this is due to the mutation of serines to aspartates having only a modest phosphomimetic effect caused by the lower negative charge of aspartic acid compared to a phosphate group. The effect of mimicking the phosphorylation of Ds is therefore no greater than the normal effect of Fj phosphorylation on Ds in the region of the wing being assayed. It is also likely that our FRAP assay is unable to detect subtle differences in binding affinity.

We also constructed Ft phosphomutants and mimics (see 'Materials and methods') and tested them in a similar manner. Consistent with previous results, a Ft phosphomutant (*Act-ft-$^{S/T>Ax5}$-EGFP*) showed a decreased stable fraction (*Figure 4C*, *Figure 4—figure supplement 1*) and a large reduction in stable amount when compared to 'wild-type' (*Figure 4C″*, 97.9 $\pm$ 15 and 194.5 $\pm$ 24 intensity units respectively, p = 0.006). Additionally, a Ft phosphomimetic (*Act-ft-$^{S/T>Dx4}$-EGFP*) showed slightly improved stability compared to 'wild-type' (*Figure 4—figure supplement 1*), however, this increase was not statistically significant. Removal of Fj, and therefore Ds phosphorylation, from the phosphomutant background (*Figure 4D*, *Figure 4—figure supplement 1*) did not result in a significant change in stable amount at the junctions confirming that the observed reduction in stability of the phosphomutant (*Figure 4C″*) was primarily due to a loss of Fj phosphorylation of Ft. A bar chart showing the stable amounts of *Act-ft-EGFP* in each condition is provided in *Figure 4F*.

Importantly, the phosphomutant experiments have revealed in vivo affects of Fj phosphorylation on both Ft and Ds. Mutating the Ds phosphorylation sites resulted in a significantly improved binding ability, however, this improved ability was lost when Ft was also not phosphorylated. Additionally, the phosphomutant experiments have confirmed that there is a dominant effect of Fj phosphorylation on Ft.

## Modelling Ft–Ds interactions with a Ft phosphorylation bias

Previously, our model of Ft–Ds interactions produced consistent cellular asymmetry with a negligible tissue gradient when Fj acted upon both Ft and Ds (*Figure 1D,E*). However, our data have shown that, although Fj can act on both Ft and Ds, the phosphorylation of Ft to increase Ft–Ds binding is dominant. Furthermore, in the absence of Ft phosphorylation (in a Ft phosphomutant), the phosphorylation of Ds ('wildtype' vs *fj$^-$*) has no significant effect on the overall amount of bound protein. In order to understand how this might affect the establishment of cellular asymmetry and binding across a tissue, we modified our model parameters as follows: first, we reduced the degree by which phosphorylated Ds inhibits Ft–Ds binding in complexes where Ft is phosphorylated (i.e., complexes A and B); second, we made the binding affinities for complexes formed from unphosphorylated Ft (i.e., complexes C and D) the same (see 'Materials and methods'). We then ran the model with Fj acting on both Ds and Ft (*Figure 5A*). We consequently still see the establishment of cellular asymmetries in Ft–Ds distribution within each cell, but also see a tissue gradient of Ft–Ds binding that follows the Fj gradient. The model therefore predicts that as you move from a region of low Fj to high Fj, Ft–Ds dimer stability and thus the stable amount at cell junctions should increase.

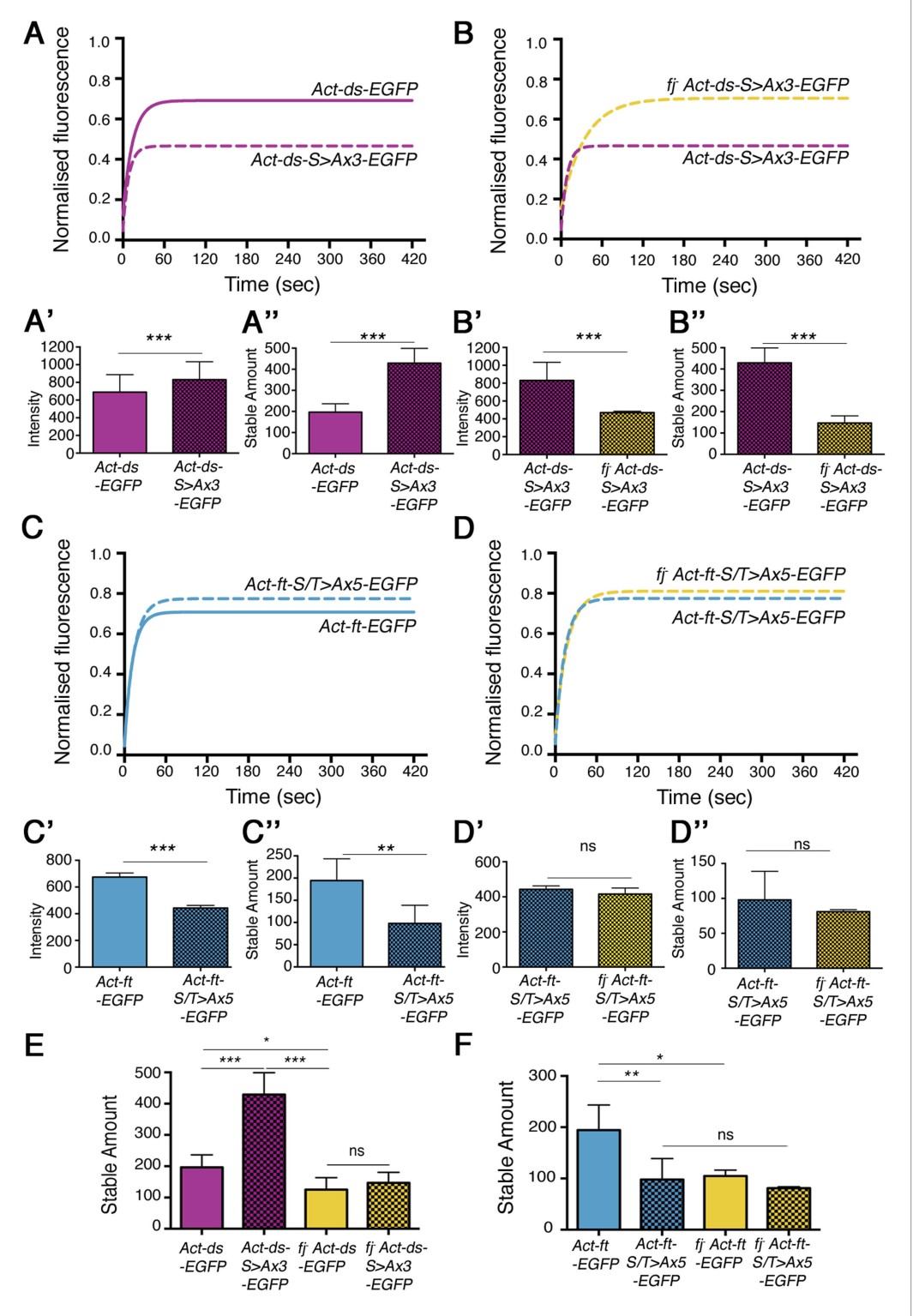

**Figure 4**. Effects of mutating the phosphorylation sites in Ds and Ft on in vivo stability. (**A**) Comparison of *Act-ds-EGFP* with *Act-ds-EGFP* phosphomutant (*Act-ds-S>Ax3-EGFP*). FRAP is performed on junctions as no puncta are visible. Ds-S>Ax3-EGFP has a larger remaining stable fraction after bleaching (**A**), is more highly expressed at junctions (**A'**) and has a significantly larger stable amount at junctions (**A"** p = 0.0004) than 'wild-type'. (**B**) When Fj is removed from the phosphomutant background (*fj⁻ Act-ds-S>Ax3-EGFP*) intensity levels drop (**B'**) and the stable amount at junctions returns to below 'wild-type' levels (**B"** and **E**). (**C**) Ft-EGFP phospho-mutant (*Act-ft-S/T>Ax5-EGFP*) has

*Figure 4. Continued*

a smaller stable fraction than Ft-EGFP (*Act-ft-EGFP*), shows reduced levels at junctions (**C′**), and has a significantly reduced stable amount at junctions (**C″**, p = 0.006). (**D, D′, D″**) In a *fj^d1* mutant background, the phosphomutant protein (*fj⁻ Act-ft-^{S/T>Ax5}-EGFP*) does not become any less stable suggesting any loss of stability in (**C**) is caused primarily by a loss of Ft phosphorylation. See *Figure 4—figure supplement 1* for FRAP graphs with individual data points. (**E–F**) An overview of stable amounts in (**E**) *Act-ds-EGFP* and (**F**) *Act-ft-EGFP* shows that despite there being a significant increase in stable Ds when its phosphorylation sites are mutated, the removal of Fj results in a loss of stable Ds (**E**; p = 0.02) and also Ft (**F**; p = 0.02). Thus the phosphorylation state of Ds only appears to significantly affect Ft–Ds binding if Ft is already phosphorylated by Fj.

The following figure supplement is available for figure 4:

**Figure supplement 1**. Individual data points for FRAP experiments.

## Ft–Ds binding varies across the Fj gradient

We next endeavoured to test in vivo the prediction of a gradient of Ft–Ds binding across the tissue. We have already shown that Fj protein levels are high in the distal wing pouch and lower proximally (*Strutt et al., 2004*; *Figure 1B*). We therefore would expect there to be an increase in stability of Ft and Ds as we move from regions of low Fj (proximal) to regions of high Fj (distal). To test this, we used FRAP analysis on endogenously expressed Ds-EGFP (*Figure 5B*, *Figure 5—figure supplement 1*) and were able to detect a statistically significant increase in stable junctional amount in more distal regions (*Figure 5B″*, 292.9 ± 19 proximally and 378 ± 18 distally, p = 0.01, 23% difference [see also *Supplementary file 1*]). This difference in stable amount was lost when Fj was removed (*Figure 5C″* 167.2 ± 22 proximally and 142.7 ± 28.6 distally, NS [see also *Supplementary file 1*]), implying that the Fj gradient normally produced the difference across the tissue.

We repeated this experiment using the *Act-ds-^{S>Ax3}-EGFP* phosphomutant, to assess the result of Fj only acting on Ft. Despite relatively small differences between stable fractions (*Figure 5E*, *Figure 5—figure supplement 1*), we saw a greater difference in stable Ds-^{S>Ax3}-EGFP at junctions between high and low Fj regions (*Figure 5E″*, 109.2 ± 21 proximally and 230.4 ± 37 distally p = 0.01, 53% difference) which was again lost upon removal of Fj (*Figure 5F″*, 97.8 ± 9 proximally and 78.2 ± 25 distally, NS). These results directly confirm the predictions of the model, showing not only that a slight gradient of binding strength exists across the tissue, but also that an opposing gradient of Ds phosphorylation usually acts to counter the effects of a gradient of Ft phosphorylation, resulting in a relatively even distribution of bound Ft–Ds complexes at junctions across the tissue axis from low to high Fj.

## Discussion

A long-standing problem in developmental biology is understanding how long-range patterning gradients are interpreted at the cellular level. More specifically, with regard to understanding how cell polarity is coordinated across sheets of cells, a major goal is to determine the mechanism by which a gradient of transcription across a tissue (produced for instance in response to a morphogen) can be sensed by individual cells to result in each cell adopting a uniform polarisation. A particular challenge for such a sensing mechanism is that the difference in levels of transcription between adjacent cells may be very small (e.g., only a few percent or less of the peak expression levels), and at the high end of the gradient this difference needs to be read against the background of a high overall expression level.

The Ft-Ds-Fj pathway in *Drosophila* represents an excellent system for addressing this problem. The Ft and Ds cadherins bind heterophilically between adjacent cells, and the visible readout of polarity is the asymmetric distribution of these dimers across the cell axis, such that approximately two-fold higher Ds is found on one cell edge, bound to Ft on the apposing edge of the neighbouring cell (*Ambegaonkar et al., 2012*; *Bosveld et al., 2012*; *Brittle et al., 2012*). However, the measured Ft asymmetry across the cell axis is weaker than that of Ds, most likely because there is a larger population of Ft present at junctions, including a presumably significant unbound fraction which is symmetrically distributed.

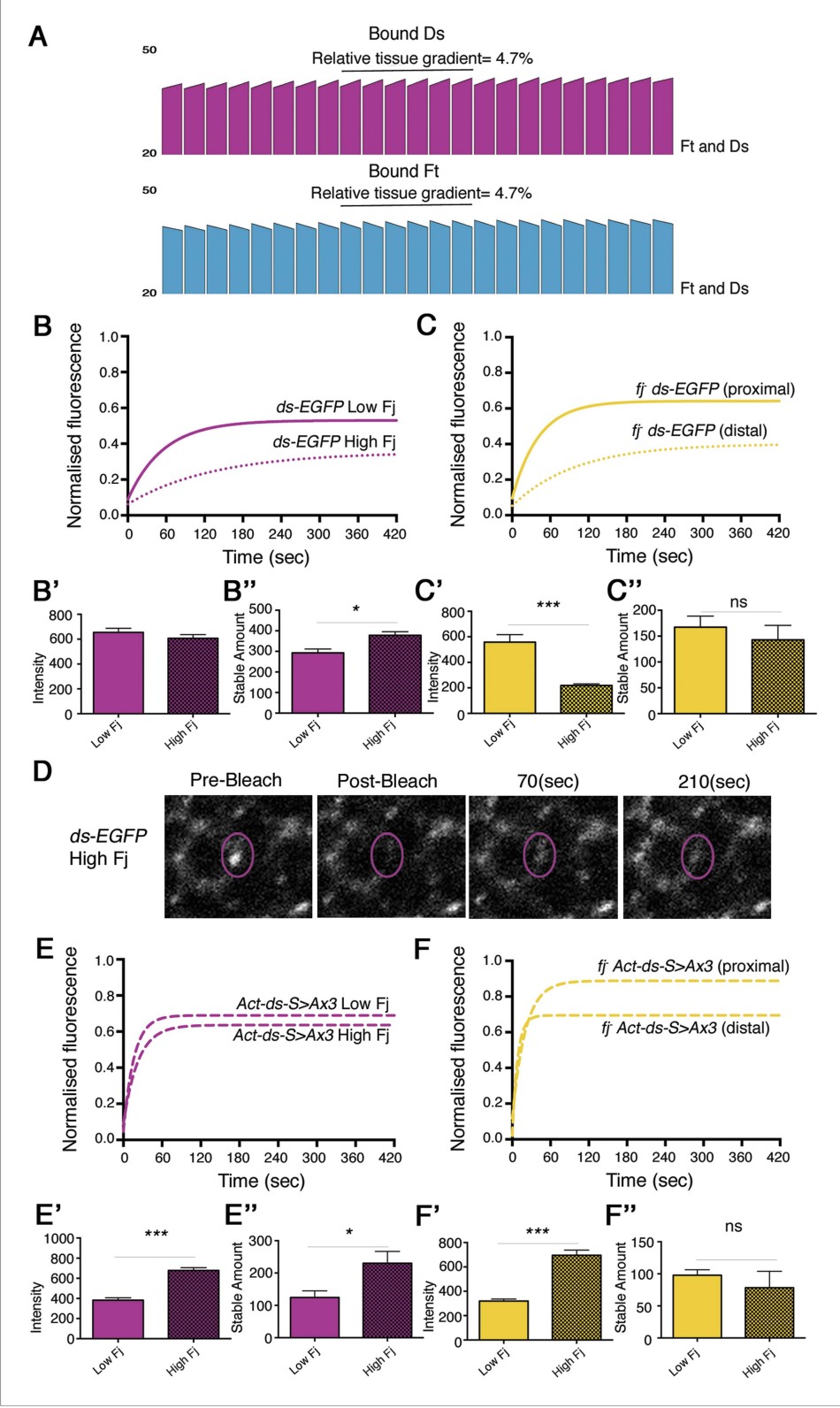

**Figure 5**. Ft–Ds binding varies across the wing in response to the Fj expression gradient. (**A**) Output of the revised computational model in which Fj acts more strongly on Ft than Ds, and in which unphosphorylated Ft binds with equal affinity to both phosphorylated and unphosphorylated Ds. When Fj can act on both Ft and Ds with a Ft bias, an intermediate outcome between acting on Ft only and acting on Ft and Ds without a bias is seen and a cellular

*Figure 5. continued on next page*

*Figure 5. Continued*

asymmetry of 4.1% is observed. A tissue gradient of 4.7% in the model when acting on both Ft and Ds, compared to 10% when Fj acts on Ft alone (*Figure 1D',E'*) suggests that Ds phosphorylation is required to counter the strong graded effects of Ft phosphorylation. (**B**) FRAP analysis of endogenously expressed Ds-EGFP reveals a larger stable fraction in regions of high Fj (distal) despite showing slightly less overall fluorescence (**B'**). Overall, a significantly increased stable amount is seen in high Fj regions (**B"** p = 0.01, 23% difference). (**C**) FRAP analysis in a *fj* mutant shows that this difference is lost when taking pre-bleach intensity levels into account (**C', C"**). See *Figure 5—figure supplement 1* for FRAP graphs with individual data points. (**D**) Images taken during a FRAP experiment in a region of high Fj demonstrate the low level of recovery over time when compared to Ds-EGFP recovery in a region of low Fj (as seen in *Figure 3C*). (**E**) FRAP analysis of junctionally localised Ds-EGFP expressed from the *Act-ds-*$^{S>Ax3}$*-EGFP* transgene show overall reduced stable fractions, however, increased distal intensity levels of Ds-EGFP (**E'**) mean the overall stable amount is significantly increased in regions of high Fj (**E"**, p = 0.01, 53% difference). In a *fj*$^{d1}$ mutant (**F**) any observed difference in stable amount across the tissue is lost (**F"**) when taking pre-bleach intensity levels into account (**F'**) (proximal equates to low Fj, distal equates to high Fj). This is again consistent with Fj acting primarily on Ft. See *Figure 5—figure supplement 1* for FRAP curves with individual data points.

The following figure supplement is available for figure 5:

**Figure supplement 1**. Individual data points for FRAP experiments.

---

Importantly, the asymmetric distribution of Ft–Ds dimers is a result of the patterns of transcription of the *ds* and *fj* genes as specified by upstream morphogens (*Ambegaonkar et al., 2012*; *Bosveld et al., 2012*; *Brittle et al., 2012*). In this study, we focus specifically on the mechanisms determining Ft–Ds subcellular polarity in the *Drosophila* third instar wing imaginal disc. At this stage, Fj is expressed in a gradient, high at the putative distal end of the wing (i.e., the centre of the wing pouch) and low proximally towards the wing hinge (*Strutt et al., 2004*), whereas Ds is relatively uniformly expressed in the visible region of the wing pouch, but higher in hinge regions of the wing (*Strutt and Strutt, 2002*). It is important to point out that much of the proximal wing blade is folded at the larval stage and most likely also has high levels of Ds expression and this region of high Ds expression could therefore also be promoting Ft–Ds asymmetry. Although we believe that both the Fj and Ds expression patterns are important cues for specifying Ft–Ds asymmetry in the wing pouch, the evidence suggests that boundaries of Ds expression may only be able to act as a patterning cue over a few cell diameters (*Ambegaonkar et al., 2012*; *Brittle et al., 2012*), and therefore throughout much of the wing pouch the Fj gradient is likely to be a dominant cue.

In in vitro assays, Fj is able to mediate the phosphorylation of extracellular cadherin repeats of Ft and Ds, and phosphorylation of Ft has been shown to promote its binding to Ds, whereas phosphorylation of Ds appears to inhibit its binding to Ft (*Ishikawa et al., 2008*; *Brittle et al., 2010*; *Simon et al., 2010*). Thus, the gradient of Fj in the wing pouch is predicted to produce opposing gradients of Ft–Ds binding affinities, with Ft–Ds binding favoured between cells moving down a Fj gradient, and Ds-Ft binding favoured between cells moving up a Fj gradient (*Figure 6A*). It has been previously proposed that these opposing binding gradients might play a role in producing uniform Ft–Ds interactions across the tissue (*Simon et al., 2010*).

In this study, we use a mixture of computational and experimental methods to address the mechanism of action of Fj. Initially, we constructed a simple one-dimensional computational model, in which Fj activity either promotes Ft binding activity or inhibits Ds binding activity, and in which Ft–Ds dimers are then allowed to freely bind at either cell edge until they reach an equilibrium state. Using this we show that a Fj gradient can lead to cellular asymmetry of Ft–Ds if Fj acts on either Ft or Ds or both. Our predictions differ from those of a previous study (*Simon et al., 2010*) which suggested that Fj acting only on Ft would not result in a cellular asymmetry of activity: the origin of this difference is that in our model (in accordance with our experimental observations) the Ft and Ds populations are mobile and able to redistribute to the most favourable cell edge. However, in agreement with the same study, our model does predict that only in the situation that Fj acts on both Ft and Ds is the amount of Ft–Ds dimers bound at the junctions approximately the same across the entire axis of the Fj gradient.

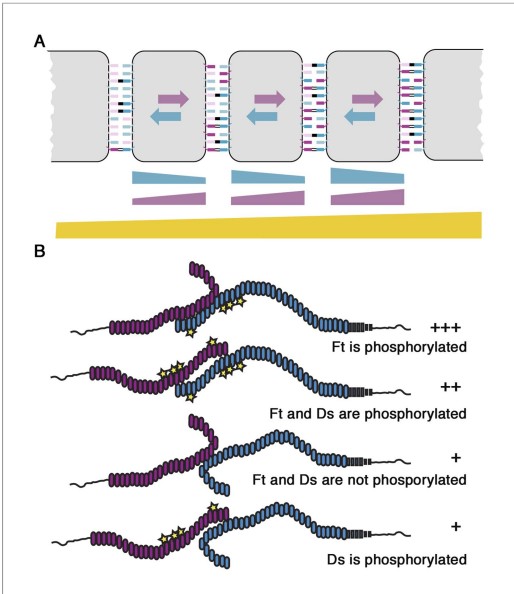

**Figure 6**. Models of Ft–Ds interactions. (**A**) Cartoon illustrating our model of Ft–Ds binding across a tissue according to a Fj gradient. Ft-P (dark blue) and Ds-P (dark purple) increase, and Ft (light blue) and Ds (light purple) decrease as you move up the Fj gradient. Weak binding (no line) between heterodimers can occur between all populations, intermediate binding (double lines) can occur when Ft-P is available and the strongest bond (solid line) occurs only between Ft-P and Ds. As there is a bias towards Ft phosphorylation increasing binding, more intermediate binding occurs in regions of high Fj even though less favourable Ds (light purple) is available. In regions of low Fj, although there is more favourable Ds available, Ft-P is less abundant meaning fewer intermediate bonds are able to form. As Ft-P allows for stronger binding in cells with higher Fj, Ds in the next left-most cell preferentially accumulates to the right cell edge meaning the Fj gradient produces cellular asymmetry. A gradient of tissue binding is also produced with cells in the left-most cell producing fewer intermediate bonds than those in the right-most cell. (**B**) Conformational changes caused by the phosphorylation of Ft and Ds could provide an explanation for the differences in binding affinity caused by apparently similar phosphorylation events. The 'open' conformation of Ft and the 'closed' conformation of Ds may bind preferably such that if Ft is phosphorylated and Ds is not, binding is strongest (+++). If Ft and Ds are both phosphorylated, binding strength is intermediate (++). 'Closed' conformation of non-phosphorylated Ft produces the weakest binding and is not affected by Ds (+).

Using FRAP to measure the levels of stable amounts and mobility of Ft and Ds at cell junctions, combined with either removal of *fj* activity or mutation of the Fj phosphorylation sites in Ft and Ds, we directly demonstrate that Ft and Ds stably associate at cell junctions in vivo, that Fj modulates this binding with a net positive effect in promoting binding, and that as predicted from the in vitro assays, Fj acts independently on both Ft and Ds to promote or inhibit their binding, respectively.

An interesting experimental observation is that the effects of Fj on Ft–Ds binding and the total amounts stably localised to cell junctions are relatively modest. For instance, blocking the phosphorylation of either Ft or Ds only decreases the stable junctional amounts by about twofold. Thus, all the molecular species (i.e., Ds and DsP and Ft and FtP) contribute to the final population of bound Ft and Ds at cell junctions.

Our experimental results reveal that the effect of Fj on Ft is stronger than the effect of Fj on Ds. A simple prediction of this observation, confirmed by our computational model, is that the ability of the gradient of phosphorylated Ds to oppose the gradient of phosphorylated Ft is reduced, and therefore a gradient of Ft–Ds binding is expected to be observed across the tissue (*Figures 5A and 6A*). This prediction was confirmed experimentally. We cannot rule out that under normal circumstances the in vivo effects of Fj on Ds are in fact negligible compared to those on Ft, as our experiments make use of Ds transgenes and may not fully represent events occurring when wild-type Ds is expressed from its endogenous locus. Nevertheless, the simplest interpretation of both our observations and previous published work is that Ds phosphorylation contributes to normal patterning. In sum, our findings demonstrate an in vivo mechanism for how a Fj expression gradient is converted into cellular asymmetries via phosphorylation of both Ds and Ft.

We note that our computational simulations result in relatively modest cellular asymmetries of Ft and Ds distributions (around 10% of the total cellular levels), whereas in vivo, observed Ds asymmetry can be as high as twofold (*Ambegaonkar et al., 2012*; *Brittle et al., 2012*). However, our model is only intended to make simple predictions regarding the effects of

binding and redistribution of Ft and Ds molecules in a simple one-dimensional system and does not capture the full complexity of a three-dimensional cell environment and changes that occur in protein production and degradation and other cell properties over time. Furthermore, it is generally believed that relatively weak asymmetries generated by expression gradients might subsequently be amplified by feedback mechanisms (*Ambegaonkar et al., 2012*; *Brittle et al., 2012*), and some

molecular mechanisms that might contribute to amplification have recently been identified (*Bosch et al., 2014*; *Rodrigues-Campos and Thompson, 2014*).

With regard to possible amplification mechanisms, an intriguing observation is that the majority of stable Ft–Ds at junctions is concentrated in bright regions which we refer to as 'puncta'. Furthermore, even mobile junctional populations of Ft and Ds show reduced mobility within these brighter regions. This might suggest that in addition to trans-interactions between Ft and Ds in neighbouring cells, there may also be cis-interactions between Ft and Ds molecules in cell junctions, a view supported by previous experimental reports (*Matakatsu and Blair, 2006*; *Sopko et al., 2009*). Such a clustering mechanism could provide the molecular basis of a positive feedback interaction. Previous observations have suggested that Ft–Ds puncta do not co-localise with 'core' planar polarity protein puncta at junctions (*Ma et al., 2003*). We are unsure as to the reasons for these distinct puncta populations, they could correspond to specialised membrane domains that favour binding or they could be random regions of protein clustering driven by cis-dimerisation or cytoskeleton tethering.

A final conundrum is the observation, first made in vitro, which we have now confirmed in vivo, that modification of analogous serine residues in cadherin repeats of Ft and Ds by Fj phosphorylation leads to opposite effects on their binding activity. The simplest model would be that phosphorylation either promoted or inhibited binding to a partner, but this is evidently not the mechanism in play here. A possible working hypothesis is that phosphorylation events lead to changes in the intramolecular conformation of the extracellular regions of Ft and Ds, by analogy to the way in which phosphorylation frequently acts to cause other classes of molecules to enter 'open' or 'closed' conformations (*Xu and Carpenter, 1999*; *Potter et al., 2005*; *Bertocchi et al., 2012*). A possible scenario is that phosphorylation of each molecule causes it to enter an 'open' conformation, and that 'open' Ft binds most favourably to Ds, but 'closed' Ds binds most favourably to Ft (*Figure 6B*). A recent publication analysing the configuration of mammalian Ft4 and Ds1 revealed multiple hairpin-like bends in their C-terminal regions caused by the loss of calcium binding linkers (*Tsukasaki et al., 2014*). It is possible that Fj phosphorylation results in conformational changes between cadherin repeat domains near to these calcium binding sites resulting in the regulation of binding strength, as suggested previously for E-cadherin (*Petrova et al., 2012*). Further studies of the structures and mode of heterophilic interactions between Ft and Ds will be required to resolve this question.

## Materials and methods

### Antibodies, immunolabelling and imaging

Wing discs were dissected from wandering third instar larvae, fixed in 4% paraformaldehyde and washed in PBS containing 0.1% Triton-X-100 prior to immunolabelling. Primary antibodies used for histology were rabbit anti-Ds (*Strutt and Strutt, 2002*), rabbit anti-Ft (*Brittle et al., 2012*), guinea pig anti-Hrs (*Lloyd et al., 2002*), mouse anti-βGAL (Promega, Wisconsin, USA), and mouse anti-Armadillo (DSHB, Iowa City, USA). A rabbit serum against Fj was generated using a fusion protein corresponding to amino acids 111–433, affinity purified and used at 1/100 for immunostaining. Secondary antibodies used were anti-Rb RRX and Cy2 (Jackson, Pennsylvania, USA), anti-guinea pig A568 and anti-mouse Cy5 (Molecular Probes, Oregon, USA). Images are averages of three confocal microscope sections taken on an Olympus FV1000 (Pennsylvania, USA), a Leica SP1 (Solms, Germany), or a Nikon A1R (Tokyo, Japan) confocal and processed using ImageJ (NIH, USA) and Adobe Photoshop (California, USA). ImageJ was used to measure the mean intensity of endogenously tagged *ds-EGFP* and *ft-EGFP* at junctions using at least nine wings of each genotype.

For Fj gradient measurements, immunolabellings were carried out on fixed wild-type wing discs using the rabbit-Fj antibody. Images were taken as 0.2-μm confocal slices throughout the disc and averages of the full stack were taken. Regions of interest were hand drawn per cell using a membrane marker as a guide and average intensity per cell was measured. Discs containing null clones were used for antibody background subtraction. Distal cells containing maximum measured signal were normalised to 100% and gradient of a proximal row of cells was calculated.

### Antibody internalisation assay

Antibody internalisation assays were carried out on 5.5 hr APF pupal wings as previously described (*Strutt et al., 2011*). To detect V5-Ft, wings from flies of the genotype $ft^{G-rv}/ft^8$; *P[acman] V5-ft* (*Feng and Irvine, 2009*) were dissected and incubated with anti-V5 antibody (Novus Biologicals, Colorado,

USA). For detection of extracellular V5-Ft, tissue was incubated in secondary antibody in the absence of detergent, and post-fixed before adding other antibodies with detergent. Internalised V5-Ft was co-stained with guinea pig anti-Hrs (*Lloyd et al., 2002*). For quantitation of extracellular staining at least five wings at each time point were imaged from at least two experiments taking 0.15-µm sections and using constant confocal settings. An average intensity of the three brightest confocal slices at the level of the apical junctions was measured in ImageJ. Laser-off background was subtracted, and the readings were normalized to 1.0 at t0.

To visualise internalised V5-Ft, wings were imaged in apical and subapical regions. Statistical analysis was carried out using ordinary one-way ANOVA with Tukey's test for multiple comparisons.

## Molecular biology

Constructs were generated using standard molecular biology techniques and mutagenised and PCR-amplified regions verified by sequencing. Full-length *ft* is a 22-kb genomic fragment from BACR11D14 (BACPAC Resources, California, USA), containing the entire coding sequence and tagged with EGFP subcloned into *pAttB-FRT-polyA-FRT* (derived from *pAct-FRT-polyA-FRT* [*Strutt, 2001*]). Point mutations in the cadherin domains were introduced using QuikChange Multi-Site Directed Mutagenesis kit (Stratagene, California, USA). 5 serines or threonines identified as phosphorylation sites in Ft (CAD3 (S273), CAD5 (S497), CAD10 (T1052), CAD11 (S1156) and CAD13 (S1387)) (*Ishikawa et al., 2008*) were mutated to alanine to generate Ft$^{-S/T>Ax5}$-EGFP. A phosphomimetic form of Ft (Ft$^{S/T>Dx4}$-EGFP) contains point mutations to aspartate of the following residues (CAD3 (S273), CAD10 (T1052), CAD11 (S1156), and CAD13 (S1387)).

## Fly strains

Alleles used are described in FlyBase. Homologous recombination was used to target EGFP into the C-terminus of the *ft* gene at its endogenous locus using the pRK2 targeting vector (*Huang et al., 2008*). *ds-EGFP* (*Brittle et al., 2012*) and *ft-EGFP* flies were recombined with $fj^{d1}$, $ft^{G-rv}$, or $ds^{UA071}$. All endogenous FRAP experiments were performed using homozygous *ds-EGFP* or *ft-EGFP*. Clones were generated using the FLP/FRT system (*Xu and Rubin, 1993*) and marked with *arm-lacZ* (*Vincent et al., 1994*). Transgenes containing *ft*-EGFP and point mutants were integrated into the same landing site (attP2 68A4) (*Groth et al., 2004*) by Genetivision (Texas, USA). *Act-ds-EGFP* and point mutations used are described previously (*Brittle et al., 2010*).

Genotypes used were:

*y w Scer\FLP1$^{Ubx.hs}$; ds$^{38k}$/ ds$^{UA071}$; AttB{w$^+$ ActP-FRT-polyA-FRT-dsX-EGFP} /+*
*y w Scer\FLP1$^{Ubx.hs}$; ds$^{UA071}$ fj$^{P1}$/ ds$^{38k}$ fj$^{P1}$; AttB{w$^+$ ActP-FRT-polyA-FRT-dsX-EGFP} /+*
*y w Scer\FLP1$^{Ubx.hs}$; ft$^8$/ ft$^{G-rv}$; AttB{w$^+$ ActP-FRT-polyA-FRT-ftX-EGFP} ft$^{G-rv}$ /+*
*y w Scer\FLP1$^{Ubx.hs}$; ft$^8$ fj$^{d1}$/ ft$^{G-rv}$ fj$^{P1}$; AttB{w$^+$ ActP-FRT-polyA-FRT-ftX-EGFP} /+*

where *dsX* and *ftX* refers to wild-type *ds* or *ft* or one of the phosphomutants.

## Live imaging and fluorescence recovery after photobleaching (FRAP)

Prior to dissection, an imaging chamber was built using a 22 × 50 mm cover glass as a base slide (Thermo Scientific, Massachusetts, USA). Sellotape was placed smoothly on to the centre of the slide and a 7-mm$^2$ area was cut out using a razor blade. Wandering stage larvae were collected and cleaned by rinsing in PBS. Wing discs were dissected in Shields and Sang M3 media (Sigma [#S3652], Missouri, USA) with 2% added foetal bovine serum (M3FBS). Discs were transferred by pipette to the cut out area in the imaging chamber with around 4 µl of media and arranged with their apical surface facing towards the base cover glass. The M3FBS was spread evenly around the cut out area. A 13-mm circular cover glass (Thermo Scientific) was carefully placed over the top allowing extra media to spread to the edges. Quick drying nail varnish was used to seal the cover glass after leaving to settle for a few minutes.

For FRAP, samples were imaged on an inverted Nikon A1R GaAsP confocal using a Nikon 60× oil objective lens at 11.76× zoom producing a FRAP region of 256 × 256 pixels with a pixel size of 0.07 µm. For pre- and post-bleach images, a 448 Argon laser was used at an output of 0.5% with varying gain settings depending upon phenotype. Eight 1 µm$^2$ regions of interest (ROIs) were selected per wing and bleached using the 488 argon laser at 50% power, passing 1–3 times for between 0.5 and 1.5 s depending upon experiment. Two pre-bleach images were captured with no delay as well as an

immediate post-bleach image, 10 images were then captured every 5 s, followed by 10 images every 10 s and 10 images every 30 s. The initial rapid imaging was done in order to capture adequate rate information. For analysis, ROIs were individually reselected in ImageJ at each time point and acquisition bleaching was measured in non-bleached regions. Data were corrected for acquisition bleaching and normalised against pre-bleach values. An XY graph was plotted for each wing in PRISM (v.6 GraphPad). A one-phase exponential association curve was fitted for each ROI and an average plateau value recorded. A one-phase exponential curve was generated as we observe a single mode of recovery that reaches a plateau. We note that should the mode of recovery be more complex than this a two-phase exponential curve may be more appropriate; however, we do not have relevant experimental data to support this. Pre-bleach values were averaged per wing and multiplied by their associated plateau giving a stable amount. Stable amounts were then averaged across wings producing a final figure for each genotype. Stable amounts were analysed using unpaired t-tests or ordinary one-way ANOVAs with Tukey's test for multiple comparisons. Final stable fraction graphs were produced using the average plot for each wing and intensity was averaged across ROIs.

## Modelling

We developed a computational model of Ft–Ds binding at cellular junctions, described by a set of ordinary differential equations. In this framework, we define a one-dimensional row of cells, whose proximal and distal membranes contain the same initial amount of Ft and Ds (*Figure 1—figure supplement 1*). These molecules were then phosphorylated according to a linear gradient of Fj activity, applied to reflect in vivo measurements of Fj quantity. Phosphorylated and unphosphorylated molecules were allowed to bind across junctions resulting in formation of four possible complexes, each in two orientations, namely, FtP—Ds (A), FtP—DsP (B), Ft—Ds (C), and Ft—DsP (D). This consideration of all four possible complexes between phosphorylated and non-phosphorylated forms of Ft and Ds is a key distinguishing feature of our model, when compared to previous computational approaches which only considered a single binding species essentially equivalent to our complex A (*Abley et al., 2013*; *Mani et al., 2013*; *Jolly et al., 2014*). Reaction rates were derived from mass action, giving the following equations for proximally oriented complexes:

$$\frac{dA_{P_i}}{dt} = ka_{on}FtP_{P_i}Ds_{D_{i-1}} - ka_{off}A_{P_i},$$

$$\frac{dB_{P_i}}{dt} = kb_{on}FtP_{P_i}DsP_{D_{i-1}} - kb_{off}B_{P_i},$$

$$\frac{dC_{P_i}}{dt} = kc_{on}Ft_{P_i}Ds_{D_{i-1}} - kc_{off}C_{P_i},$$

$$\frac{dD_{P_i}}{dt} = kd_{on}Ft_{P_i}DsP_{D_{i-1}} - kd_{off}D_{P_i}.$$

Subscripts denote the proximal (P) membrane in cell *i* neighbouring distal membrane (D) in cell *i−1*. Equivalent equations were derived for distally oriented complexes. Each reaction is parameterised by rate constants $k_{on}$ and $k_{off}$ for binding and unbinding reactions of each complex. Equations for unbound molecules were derived in a similar fashion, with the addition of a simple term allowing redistribution within a cell, parameterised by the coefficient, *Diff*:

$$\frac{dDs_{P_i}}{dt} = -ka_{on}FtP_{D_{i-1}}Ds_{P_i} - kc_{on}Ft_{D_{i-1}}Ds_{P_i} + ka_{off}A_{D_{i-1}} + kc_{off}C_{D_{i-1}} + Diff(Ds_{D_i} - Ds_{P_i}),$$

$$\frac{dDsP_{P_i}}{dt} = -kb_{on}FtP_{D_{i-1}}DsP_{P_i} - kd_{on}Ft_{D_{i-1}}DsP_{P_i} + kb_{off}B_{D_{i-1}} + kd_{off}D_{D_{i-1}} + Diff(DsP_{D_i} - DsP_{P_i}),$$

$$\frac{dFt_{P_i}}{dt} = -kc_{on}Ft_{P_i}Ds_{D_{i-1}} - kd_{on}Ft_{P_i}DsP_{D_{i-1}} + kc_{off}C_{P_i} + kd_{off}D_{P_i} + Diff(Ft_{D_i} - Ft_{P_i}),$$

$$\frac{dFtP_{P_i}}{dt} = -ka_{on}FtP_{P_i}Ds_{D_{i-1}} - kb_{on}FtP_{P_i}DsP_{D_{i-1}} + ka_{off}A_{P_i} + kb_{off}B_{P_i} + Diff(FtP_{D_i} - FtP_{P_i}).$$

Binding rates between molecules are parameterised by the association constant ($k_{on}/k_{off}$) for each complex, since a higher 'on' rate constant will result in an increased concentration of complexed molecules. Relative binding strengths of different molecule combinations reflect findings from in vivo and in vitro studies (*Brittle et al., 2010*; *Simon et al., 2010*), such that phosphorylation of Ft (FtP) promotes its binding and conversely phosphorylation of Ds (DsP) inhibits its binding. The following hierarchy of relevant association constants was used initially, A > B = C > D, such that the complex containing the two favoured molecules, FtP and Ds making complex A, had a faster 'on' rate than other combinations. Initial values of $k_{on}/k_{off}$ for A, B, C, and D were chosen as 1, 1/4, 1/4 and 1/16, respectively, to reflect the relative differences in stable amounts measured experimentally.

Binding was allowed to continue over time until convergence. Simulations were run using an in-built ode solver (ode23s) in MATLAB (R2013a; MathWorks, Massachusetts, USA) and final concentrations of bound molecules were plotted.

We went on to adapt this model to reflect our experimental finding that Fj appears to have a dominant effect on Ft. To achieve this, we altered association constants of binding reactions to allow phosphorylation of Ds to have a less significant effect on binding strengths, giving A > B > C = D. Thus association constants for each complex A, B, C, and D, were given as 1, 1/2, 1/4, and 1/4, respectively. Further explanation can be found in the 'Results'.

## Acknowledgements

We thank Ken Irvine for the *P[acman]V5-ft* fly stock, Hugo Bellen for the anti-Hrs antibody, John Walker for generating the *ft-EGFP* stock, and Katrina Hofstra for assistance producing the anti-Fj antibody. DNA clones were obtained from BacPac Resources and antibodies from the DSHB. This work was supported by a Wellcome Trust Senior Fellowship to DS Confocal facilities were provided by the Wellcome Trust, MRC and Yorkshire Cancer Research.

## Additional information

### Funding

| Funder | Grant reference number | Author |
|---|---|---|
| Wellcome Trust | 100986 | David Strutt |
| Wellcome Trust | 089717 | Amy L Brittle, David Strutt |

The funders had no role in study design, data collection and interpretation, or the decision to submit the work for publication.

### Author contributions

RH, ALB, Conception and design, Acquisition of data, Analysis and interpretation of data, Drafting or revising the article; KHF, Conception and design, Acquisition of data, Analysis and interpretation of data; NAMM, Conception and design; DS, Conception and design, Drafting or revising the article

## Additional files

### Supplementary file

• Supplementary file 1. Table contains numbers of wings, plateau data, and 95% confidence of plateau data for each FRAP experiment.

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
