## [Decision Letter]

Thank you for sending your work entitled “Interpreting a long-range gradient at the cellular level: the mechanism of action of Four-jointed in the *Drosophila* wing” for consideration at *eLife*. Your article has been evaluated by K VijayRaghavan (Senior editor), and four reviewers, one of whom, Peter Lawrence, is serving as a guest Reviewing editor.

The Reviewing editor and the other reviewers discussed their comments before we reached this decision, and the Reviewing editor has assembled the following comments to help you prepare a revised submission.

I have now assembled four reviews and all agree that a revised version of your paper will be published in *eLife*. We are impressed with the directness of your approach to an important and challenging problem that, up to now, most have preferred to simulate rather than investigate. No additional experiments are asked for but taken together there are a large number of criticisms that in my view arise mainly from a lack of clarity in your manuscript. The techniques you use, the way you present the data and some of your descriptions are complicated, and have caused confusion. So I think the simplest way to proceed is for me to send you these criticisms more or less undiluted.

The only suggestion we all agree on is that you should provide more details of your theoretical model, its assumptions, equations, and parameters so readers can understand it better and compare it with other similar models. There are other points that come up in more than one review. But my opinion is based on the fact that your paper is yours, not ours, your names will go forward with the paper and therefore I believe you, not us, should decide how to revise it. But I urge you to consider the points raised, try to make it clearer. In this case I have decided *not* to ask you to provide a reasoned explanation for all you have decided to do and not to do. This would take a huge amount of time and effort and the result might well be longer than the paper and written only really to satisfy the curiosity of we reviewers. However I know that as a responsible scientist you will want to take these criticisms seriously and do your best to make your paper as lucid as you can.

Regarding the Jolly model, if you go to PLoS ONE you will see that we already raised a query with Jolly about their model.

Here is a list of all the comments for you to consider:

Reviewer 1:

This is an interesting and accomplished article. It addresses in a new and concrete way a long-standing question that asks how a shallow multicellular gradient can be converted into a consistently oriented and asymmetric distribution of molecules within single cells. This question has often been explored by modelling but the models are not usually checked out by observation. However this paper is based on the Ft/Ds system for planar cell polarity and combines molecular measurements with modelling. The amount of bound Ds and Ft is estimated by an apparently ingenious use of FRAP. The paper has interesting findings: for the first time the authors have weighed the relative contributions of the Ds, Ft and Fj distributions to the final pattern and tried to quantify how and why the effects of Fj on Ds and Ft differ in strength. They have found evidence for a central tenet of the Ds/Ft model, as proposed earlier, that there should be a gradient of Ds/Ft heterodimers across the tissue. They have estimated the steepness of the Fj gradient from measurements based on Fj antibody. In our opinion the paper clearly deserves publication in *eLife* but only after revision, the main purpose of which would be to make its scientific strategy and results clearer for the reader.

Results:

There are problems with the Results section in that the details of some experiments confuse us, possibly because the authors have not presented their results as clearly as they should. Below we list some of the problems we have encountered in trying to understand both the data and the model. The results of the authors' model are presented in Figure 1. We have tried to work out how the model is set up and what the parameters, rules and equations are (as shown in the first paragraph), but have not succeeded. Perhaps more could be explained in a legend or in the Methods?

In the first paragraph of the Results section, the expression “a hierarchy of binding affinities”: what does this mean in terms of the model and how is this hierarchy calculated?

Fourth paragraph of the Results: “the speed… of the mobile population” ?

Figure 2 are not explained well, e.g. in the fifth paragraph of the Results section, the unstable fraction is estimated by a height of the curve, while the stable fraction is the area above the line, we can work out how the area is calculated but it is not well explained how a height is compared with an area.

In making these exponential curves the data is “fitted” to the curve but it is not clear how this is done, how objective it is, how good the fit is. After all the data could also be “fitted” to a straight line; presumably the fit would not be so good.

Seventh paragraph of the Results section onwards: it might be better to tell us at the beginning, that in ft- and in ds-, no puncta are seen (although we are not sure if that is so of ds-?)

Ninth paragraph of the Results section: this is an interesting finding and consequently we ask has the excess of Ft over Ds been put in the model?

Eleventh and twelfth paragraphs of the Results section and Figure 3: this is a difficult section, particularly as the effects of Fj on Ds are going to be declared insignificant, when compared to the effects of Fj on Ft, and therefore one is led down two paths and one seems to go nowhere. It is hard on the brain and I think the authors should do more work to make the data behind Figure 3 more accessible. For example consider Figure 3 and Figure 3. Are these results in accord with Figure 2?

Thirteenth and fourteenth paragraphs of the Results section: This is also a difficult section as one is trying to deduce the effects on heterodimers from the effect of Fj on Ft or Ds in conditions in which one of these effects is blocked by a mutation in the site of Fj's action. We get the impression that the results again show that the effect of Fj on Ds is insignificant. If this is so, then couldn't the whole be presented more simply and the effect on Fj ignored in these presentations (there is still evidence in vivo published in Brittle et al. that Fj can effect Ds ability to bind to Ft, but this was with overexpression of Fj, so this matter should be addressed in the Discussion). Figure 4 would seem to show almost no effect, barely at the level of significance. If this is so, then the headline for the thirteenth paragraph of the Results section and the statement in the eighteenth paragraph could mislead us?

The histograms in Figure 4, we are confused by them. They do show large effects, but are these differences caused more by the original intensities of levels in the puncta and non-puncta than by the variable that is pertinent (which is the difference in stability due to phosphorylation)? Consider Figure 4 for instance where we seem to be misled by the p=0.006. An example to explain (if we understand correctly): if you have an intensity of 1000 in the puncta and a recovery of 0.8 it gives 800 as a stable amount. If in the non-puncta there is an intensity of 200 and a recovery also of 0.8 the result would be 160; and course the difference between 160 and 800 is great but those two figures are not the figures one should be comparing as the recovery is 0.8 in both cases.

Another example for illustration: imagine you have, in the puncta, an intensity of 1000 and a recovery of 0.4, giving a stable amount of 400 and in the non-puncta an intensity of 200 and a recovery of 0.9, giving a stable amount of 180. In this case 400 is clearly higher than 180, but can we not be misled if we use these figures? The differences between the two fitted curves are really what matters here, is that not so? Therefore should the histograms be left out as they confuse and give an impression of a difference that is not the difference that matters?

In the legend of Figure 2, calculation of “stable amount” could be better explained and does this data fit with results shown in Figure 5.

Thinking about Figure 4 and the disparity of the effects of Fj on Ds and Ft: the effects measured would depend on locality in the gradient (Figure 5). Indeed in vivo results depended on where the clone was made (Brittle et al.). And we see here the effect of Fj varies across the wing. Thus where in the wing these measurements are made is likely to be an important factor but it is not controlled here?

Where Fj is high shouldn't Ds be low, therefore if Fj is removed, stability should surely be greater where Fj is normally high, not as in Figure 4 where we see the reverse?

Does Figure 4 show a significant change, i.e. a significant effect on Ds?

Eighteenth paragraph of the Results section: We don't know the parameters used to set up the model, a problem for the reader of this section.

Paragraph 19 of the Results section: where are these measurements made, puncta or non-puncta?

Figure 6: the model presented in A is virtually identical to that presented in [8], is that not so?

It might be useful for the authors to consider the possibility that phosphorylation may not be all or none (per Ds or Ft molecule) but could be partial. Would that possibility change interpretations?

Reviewer 2:

The authors are attempting to understand the long-standing developmental biology problem of how global tissue wide patterning gradients can lead to local cellular asymmetries of protein interactions and localization. This is a highly important question that should be of interest to a wide range of audiences. The Fat-Dachsous-Four-jointed pathway in *Drosophila* presents an ideal system to investigate the mechanisms of planar cell polarity establishment. The authors are attempting to understand the problem quantitatively by developing a 1D computational model. Using experimentally measured values for a Four-jointed (Fj) gradient, and relative binding affinities between phosphorylated and unphosphorylated Fat and Dachsous, they claim to be able to computational explain the observed in vivo patterns of Fat-Dachsous binding and planar polarization across the wing.

Major concern:

The attempt to measure in vivo protein stabilities and deduce relative binding affinities is to be applauded. However, even with these new in vivo parameters, their model still did not produce in vivo like cellular asymmetries of Fat and Dachsous distribution, which is their main readout for the model. The main interest in this subject area is how relatively mild patterning gradients across tissues can be interpreted to give relatively high cellular asymmetries, such as the 2-fold differences in Ds that has been experimentally measured previously. I feel the authors have not been able to address this issue fully, and merely show a mild cellular asymmetry as a result of their new proposed model, which is only slightly better than their first model, where they did not have the new in vivo parameters. Admittedly, the authors do acknowledge this problem, and discuss several possible mechanisms for generating a higher cellular asymmetry, such as feedback and amplification mechanisms. Unless the authors can attempt to include these mechanisms and show that they can be significant factors in generating the high cellular asymmetries, the mystery, in my opinion, still exists. As a result, in its current state, this work does not represent a significant enough advance on the field to be suitable for publication in *eLife*.

Other points to address:

1) The authors only measure the Fj gradient in one quadrant of the wing pouch (Figure 1). What about the other regions (quadrants) of the wing pouch? What are the gradients of Fj? If these are used in the model, can these gradients produce the same asymmetries in Fat and Ds (using same parameters for everything else)?

2) To go with Figure 2, please provide images to show what is a 'puncta' bleach and a 'non-puncta' bleach. It is hard to see how one would define and select these regions.

3) For Figures 1 and 5 it would be useful to show the actual values of the bound Ds and bound Ft on either side of each cell. The current graphs can only show asymmetries, but not how much, and whether that changes across the tissue. For example, for bound Ds in Figure 5, the asymmetry seems greater more distally in the tissue. Is that true in vivo?

4) The authors should provide more details for the set-up of the model, such as the equations for the ODE and actual parameter values.

Reviewer 3:

The manuscript is an informative and valuable extension of the authors' 2010 study on the role of Fj in the regulation of Ft-Ds interactions, informed by more recent findings on the polarization of Ds and Ft localization within single cells. The former study was largely limited to measurements of Ft-Ds binding in vitro, and the current work extends this to in vivo work using late third instar wing imaginal discs.

The work looks pretty straightforward, although the authors have had to make some assumption about the relationship between stability, recovery and binding; it is always possible that some of the changes are caused by something other than Ft-Ds binding, such as trafficking or protein stability. Nonetheless, the results largely match the expectations based on previous work, with only a few exceptions. These include the finding that the Fj effect on Ft seems to predominate in vivo over its effects on Ds, and that this creates a slight gradient of Ft-Ds binding strength across the wing that depends on distal Fj. The authors also find that a phosphomimetic version of Ds does not show the expected decrease in stable binding. The authors argue that the mutations do not perfectly mimic phosphorylation, and indeed that matches the reduced effect this version had in their previous in vitro assays, where they made the same argument.

Other than the concerns about less direct causes of the results, I had only a couple major comments:

1) The authors' data indicates that Ft-Ds binding is less stable in the absence of Fj. The one counter-example I found in the literature was rather less direct, but in Matakatsu and Blair (2012) the Hippo signaling boundary effects created by Ft overexpression extended further distally in the absence of Fj than its presence, suggesting stronger Ft-Ds binding in the absence of Fj. Is it possible that the result depends as well on the levels of Ft or Ds being expressed? That might make one a bit cautious about results at anything other than endogenous expression levels.

2) If the authors think that the polarization of Ft, Ds and Dachs in the distal wing/pouch is due entirely to Fj, has anyone demonstrated that? [6] only looked at dachs in *fj* ds double mutants, I believe. Bosveld looked at *fj* mutants but in the notum, not wing, and to my eye there was still some Ds polarization.

Reviewer 4:

This manuscript focuses on how long range gradients contribute to planar cell polarity. I focused primarily on assessing the modeling and FRAP analysis used in this work. In general, the interpretation of the FRAP data is consistent with the conceptual model presented by the authors, and the experiments appear to be well performed and thorough. However, I have several concerns regarding the computational model. Other points also require further clarification.

1) The details of the mathematical model are not presented here. This makes it almost impossible to evaluate the soundness of the model. The authors are advised to present the complete underlying assumptions and governing PDE equations, boundary conditions, initial conditions, and the parameters used for the modeling. Some of these parameters (such as the diffusion coefficients of Ft and Ds) could be extracted from the experiments presented. In addition, the authors indicate that model is in PDE (diffusion was included), but the numerical scheme presented here was determined using an ODE solver. This point needs to be clarified.

2) The way the kinetic equations (hierarchy of relevant associations) are presented in the Method is confusing. They should be represented using standard notations.

3) The manuscript fails to cite and discuss a recent study using mathematical modeling to examine mechanisms contributing to the sub-cellular asymmetry of Ft-Ds heterodimers ([16], Mathematical Modeling of Sub-Cellular Asymmetry of Fat-Dachsous Heterodimer for Generation of Planar Cell Polarity, PLoS ONE, 9(5):e97641). This is very surprising given that the work of Jolly et al. addresses very similar questions as the current study, albeit solely from a theoretical perspective. It is essential that the authors elaborate on how their current findings relate to this published work.

4) In this study, FRAP is used to monitor changes in putative interactions between proteins that occur between two cells and thus in separate membranes. Under these specialized conditions, one might intuitively expect diffusion of both binding partners to be slowed, as the authors suggest. However, it seems likely that other types of interactions would be required to cause immobilization per se. The authors suggest that cis-interactions may be important (eighth paragraph of the Discussion section). Other types of events, such as interactions of proteins with the actin cytoskeleton might be important as an immobilization mechanism and/or the establishment of “stable” populations. It appears that such a mechanism was previously suggested to be important for the case of E-cadherin in puncta (Cavey et al., Nature, 2008, 453:751-6, A two-tiered mechanism for stabilization and immobilization of E-cadherin). This possibility should be discussed.

5) On a related note, the authors report that Ft and Ds localize in part to puncta where the proteins appear to be immobilized but do not say very much about the possible identity of the puncta or their relationship to other puncta previously identified. These points should be further developed in the Discussion.

6) The authors use the word “stable” in several different contexts, beginning in the second headline of the Results section. For the case of the FRAP measurements, they use this to refer to an immobile fraction of proteins. They use it again to refer to a pool of proteins that do not undergo internalization in the endocytosis assay. These two “stable” pools are not necessarily identical. As such, the exact meaning of “stable” needs to be better qualified in each case.

---

## [Author Response]

We have now revised our manuscript in response to the comments from the four expert reviewers. The key comments were regarding general clarity in the main text, particularly regarding our analysis and interpretation of FRAP experiments, as well as requesting more detail on the model set up and equations. We have addressed these concerns in our revised draft, adding three new supplementary figures as well as more detailed explanations to aid understanding and clarity. Further specific points are addressed individually below.

Reviewer 1:

*There are problems with the Results section in that the details of some experiments confuse us, possibly because the authors have not presented their results as clearly as they should. Below we list some of the problems we have encountered in trying to understand both the data and the model. The results of the authors' model are presented in*
Figure 1*. We have tried to work out how the model is set up and what the parameters, rules and equations are (as shown in the first paragraph), but have not succeeded. Perhaps more could be explained in a legend or in the Methods?*

In the first paragraph of the Results section, the expression “a hierarchy of binding affinities”: what does this mean in terms of the model and how is this hierarchy calculated?

Further explanation has been added to the Methods. In short, the “hierarchy” reflects the previously published findings that FtP binds more strongly than Ft and DsP binds less strongly than Ds (7; 28). A complex that contains both favoured molecules (complex A with FtP and Ds) will bind better than one which contains both least favoured molecules (complex D with Ft and DsP). The remaining combinations of molecules making up complexes B and C will bind somewhere in the middle. We have given B and C association constants and began by making them equal to one another since we had no evidence to suggest one was better at binding than the other. The values broadly reflect the differences in stable amount we have measured in our experiments. For example, complex A and C both contain unphosphorylated Ds, but differ in the phosphorylation of Ft. We have set association constants as 1 and ¼ for A and C respectively, which is similar to the differences in stable amount seen in Figure 4 comparing Ds phosphomutant in the presence (complex A) or absence (complex C) of Fj.

Fourth paragraph of the Results section: “the speed… of the mobile population”?

We have extensively rewritten this part of the text. In particular, see additional text in the section “Ft and Ds exhibit stable populations at the cell junctions” beginning: “Additionally, if the speed of protein recovery…”

Figure 2
*are not explained well, e.g. in the fifth paragraph of the Results section, the unstable fraction is estimated by a height of the curve, while the stable fraction is the area above the line, we can work out how the area is calculated but it is not well explained how a height is compared with an area*.

Further explanation of FRAP and its interpretation has been added to the section headed “Ft and Ds exhibit stable populations at the cell junctions”. A supplementary figure has also been added explaining this point. In essence our statement “area above the line” was confusing: it is in fact the height above the line which is important.

*In making these exponential curves the data is “fitted” to the curve but it is not clear how this is done, how objective it is, how good the fit is. After all the data could also be “fitted” to a straight line; presumably the fit would not be so good*.

All FRAP data points and curves have been added to Figure 2—figure supplement 1, Figure 2—figure supplement 2, Figure 2—figure supplement 3, Figure 2—figure supplement 4 and Figure 2—figure supplement 5, along with the 95% confidence intervals of the curve fit in Table 1. This illustrates how a one-phase exponential curve fits very well in each case.

Seventh paragraph of the Results section onwards: it might be better to tell us at the beginning, that in ft- and in ds-, no puncta are seen (although we are not sure if that is so of ds-?)

This has been added (see the sentence beginning with “Removal of the putative binding partner…”, in the subsection “Ft and Ds require each other for junctional stability”, in Results).

*Ninth paragraph of the Results section*: *this is an interesting finding and consequently we ask has the excess of Ft over Ds been put in the model?*

Simply adding a 2-fold excess of Ft into our current model has only a moderate effect on cellular asymmetry measured in terms of bound Ft and Ds (however the “excess” unbound Ft does of course obscure observation of the bound fraction at junctions, hence the need to measure it using FRAP). A 10x increase in Ft still only moderately alters cellular asymmetry (from 7.8% to 6.5%). We plan to publish a full exploration of the model elsewhere, which will include changes in levels.

*The histograms in*
Figure 4*, we are confused by them. They do show large effects, but are these differences caused more by the original intensities of levels in the puncta and non-puncta than by the variable that is pertinent (which is the difference in stability due to phosphorylation)? Consider*
Figure 4
*for instance where we seem to be misled by the p=0.006. An example to explain (if we understand correctly): if you have an intensity of 1000 in the puncta and a recovery of 0.8 it gives 800 as a stable amount. If in the non-puncta there is an intensity of 200 and a recovery also of 0.8 the result would be 160; and course the difference between 160 and 800 is great but those two figures are not the figures one should be comparing as the recovery is 0.8 in both cases*.

Another example for illustration: imagine you have, in the puncta, an intensity of 1000 and a recovery of 0.4, giving a stable amount of 400 and in the non-puncta an intensity of 200 and a recovery of 0.9, giving a stable amount of 180. In this case 400 is clearly higher than 180, but can we not be misled if we use these figures? The differences between the two fitted curves are really what matters here, is that not so? Therefore should the histograms be left out as they confuse and give an impression of a difference that is not the difference that matters?

The FRAP experiments in Figure 4 are carried out on transgenic proteins under the control of the actin promoter and are thus expressed at high levels with no distinguishable puncta. We have added the sentence “FRAP is performed on junctions as no puncta are visible” into the figure legend for Figure 4.

Since we are comparing two different fluorescent proteins, or indeed in other figures we are comparing different genetic backgrounds, we must take their differing expression levels into account. We are using FRAP to infer the amount of protein stably localised (and thus presumed bound) at junctions and comparing this in different conditions. If the recovery is 0.8 in two different conditions, this is in fact not the value we are interested in, since we want to get an idea of the amount of stable protein present in the junction. We have added Figure 1 to clarify this.

*In the legend of*
Figure 2*, calculation of “stable amount” could be better explained and does this data fit with results shown in*
Figure 5
*and C'*.

*Thinking about*
Figure 4
*and the disparity of the effects of Fj on Ds and Ft: the effects measured would depend on locality in the gradient (*Figure 5*). Indeed in vivo results depended on where the clone was made (Brittle et al.). And we see here the effect of Fj varies across the wing. Thus where in the wing these measurements are made is likely to be an important factor but it is not controlled here?*

FRAP experiments are always carried out in the same region of the wing for this reason and are thus controlled. This has now been made clear in the text.

It might be useful for the authors to consider the possibility that phosphorylation may not be all or none (per Ds or Ft molecule) but could be partial. Would that possibility change interpretations?

This is an interesting idea and in terms of the model could significantly increase the number of possible complexes we need to consider. For example, if Ds has three phosphorylation sites and we look at all possible combinations of none, 1, 2 or 3 sites being phosphorylated, there are 8 possible Ds configurations. Since we don’t believe that this would change the outcomes of the model and there is no experimental evidence to give us an idea of binding strengths of these molecules in various combinations, this is beyond the scope of our current work and not adding value given the degree of uncertainty involved. We have added a discussion point to our manuscript (see the sentence beginning with “An interesting experimental observation…”, in the Discussion).

Reviewer 2:

*1) The authors only measure the Fj gradient in one quadrant of the wing pouch (*Figure 1*). What about the other regions (quadrants) of the wing pouch? What are the gradients of Fj? If these are used in the model*, *can these gradients produce the same asymmetries in Fat and Ds (using same parameters for everything else)?*

Our study attempts to address the general question of how a gradient can be turned into a cellular asymmetry rather than a study of the wing overall. Thus we have chosen to use a region where the asymmetries are clear in order to approach this issue. Other areas of the wing clearly have different asymmetries/gradients, but are usually weaker and thus would give less robust results. In the modeling framework a steeper gradient does give a stronger cellular asymmetry as one might expect.

Reviewer 3:

*1) The authors' data indicates that Ft-Ds binding is less stable in the absence of Fj. The one counter-example I found in the literature was rather less direct, but in Matakatsu and Blair (2012) the Hippo signaling boundary effects created by Ft overexpression extended further distally in the absence of Fj than its presence*, *suggesting stronger Ft-Ds binding in the absence of Fj. Is it possible that the result depends as well on the levels of Ft or Ds being expressed? That might make one a bit cautious about results at anything other than endogenous expression levels.*

Ft overexpression in clones will recruit Ds from neighbouring wildtype cells thus causing inversions on the distal side of clones by relocalising Ds to proximal cell edges. This relocalisation of Ds is against the effect of the Fj gradient (which is promoting distal localisation of Ds), so removal of Fj will increase Ft-Ds asymmetry in tissue on the distal side of Ft overexpression clones: see for instance the increased non-autonomy around Ft and Ds over-expression clones in a *fj* background in the abdomen in [8]. We therefore infer that increased Ft asymmetry, rather than stability of binding, is the driving factor for Dachs regulation and Hippo activation.

*2) If the authors think that the polarization of Ft, Ds and Dachs in the distal wing/pouch is due entirely to Fj, has anyone demonstrated that?*
[6]
*only looked at dachs in* fj *ds double mutants, I believe. Bosveld looked at* fj *mutants but in the notum, not wing, and to my eye there was still some Ds polarization*.

In the proximal wing we agree that Fj is not the only mechanism, in part (as pointed out by the reviewer) because we've previously reported residual Dachs asymmetry in a *fj* background in this region. In more distal regions we do not have any definitive data. Therefore we have expanded our discussion to reinforce the view that a Ds boundary may also be important, particularly in the proximal wing.

Reviewer 4:

*In this study, FRAP is used to monitor changes in putative interactions between proteins that occur between two cells and thus in separate membranes. Under these specialized conditions, one might intuitively expect diffusion of both binding partners to be slowed, as the authors suggest. However, it seems likely that other types of interactions would be required to cause immobilization per se. The authors suggest that cis interactions may be important (eighth paragraph of the Discussion section). Other types of events, such as interactions of proteins with the actin cytoskeleton might be important as an immobilization mechanism and/or the establishment of “stable” populations. It appears that such a mechanism was previously suggested to be important for the case of E-cadherin in puncta (Cavey et al., Nature, 2008, 453:751-6, A two-tiered mechanism for stabilization and immobilization of E-cadherin). This possibility should be discussed*.

*On a related note, the authors report that Ft and Ds localize in part to puncta where the proteins appear to be immobilized but do not say very much about the possible identity of the puncta or their relationship to other puncta previously identified. These points should be further developed in the Discussion*.

This is an intriguing point and we have expanded our discussion of possible origins of puncta and how they differ from other puncta observed at cell junctions (see the sentence, in the Results, beginning with “Additionally, mechanisms such as…”, as well as the sentence, in the Discussion, “Previous observations have suggested that Ft-Ds puncta…”.

We were also asked to include the few items of data originally referred to as “data not shown” (which we omitted in order to avoid a long Supplemental Data section that might try the patience of readers). This is now in Figure 3.

A number of other comments were made by reviewers regarding the mathematical model presented here. We would like to clarify that the model was used to simulate different possible scenarios of the Fj mechanism, giving qualitative outcomes of Ft-Ds binding. Several other models have been previously published to investigate Ft-Ds polarity and Reviewer 4 specifically mentions [16]. Other papers also include Mani et al., PNAS, 2012 and Abley et al., Development, 2013, all three of which only consider one possible complex forming, essentially equivalent in our model to complex A (FtP–Ds). This means that in each study the authors needed to implement feedback mechanisms to avoid non-uniform levels of bound material across the tissue. We have added citations to all three publications and a description of this key difference (see Materials and methods). Reviewer 2 noticed subtle differences in the modelled cellular asymmetries across the tissue presented in Figure 5. Upon further examination we found that this was an artifact of the presentation of the model implementation, and we have thus replaced the figure, which we hope appears clearer. Regardless, there is no significant difference in asymmetries across the tissue in any of our modeling figures, furthermore, we now indicate the average cellular asymmetry for each graph in the figure legends. There was some confusion over whether this was an ODE or PDE model. Since we have stated that we included a diffusion term in our equations, one might presume that we are considering how molecule concentrations vary over space as well as time, thus requiring PDEs. We have now renamed this a redistribution term, since we have simply redistributed unbound molecules across the two membranes of each cell within our ODE model.

Finally, other mechanisms such as amplification, production/degradation and trafficking have not been explored in the current version of the model. Also, since this is not presented as a modelling paper, we have not presented a full exploration of parameter space. We plan to address these issues in a separate theoretical work at a later date.